# Identification of human progenitors of exhausted CD8+ T cells associated with elevated IFN-γ response in early phase of viral infection

Curtis Cai [1,2,9], Jerome Samir[1,2,9], Mehdi R. Pirozyan[1,2], Thiruni N. Adikari[1,2], Money Gupta[1,2], Preston Leung[1,2], Brendan Hughes [1,2], Willem Van der Byl [1,2], Simone Rizzetto [1,2], Auda Elthala[2,3], Elizabeth Keoshkerian[1,2], Jean-Louis Palgen[1,2], Timothy Peters [4], Thi H. O. Nguyen [5], Raymond Louie [1,2], Katherine Kedzierska [5], Silvana Gaudieri[6,7,8], Rowena A. Bull [1,2], Andrew R. Lloyd [2] & Fabio Luciani [1,2,4] ✉

T cell exhaustion is a hallmark of hepatitis C virus (HCV) infection and limits protective immunity in chronic viral infections and cancer. Limited knowledge exists of the initial viral and immune dynamics that characterise exhaustion in humans. We studied longitudinal blood samples from a unique cohort of individuals with primary infection using single-cell multi-omics to identify the functions and phenotypes of HCV-specific CD8+ T cells. Early elevated IFN-γ response against the transmitted virus is associated with the rate of immune escape, larger clonal expansion, and early onset of exhaustion. Irrespective of disease outcome, we find heterogeneous subsets of progenitors of exhaustion, based on the level of PD-1 expression and loss of AP-1 transcription factors. Intra-clonal analysis shows distinct trajectories with multiple fates and evolutionary plasticity of precursor cells. These findings challenge the current paradigm on the contribution of CD8+ T cells to HCV disease outcome and provide data for future studies on T cell differentiation in human infections.

The cytotoxic CD8+ T cell response during primary viral infection involves recruitment, expansion, and differentiation of epitope-specific clones from the naïve population. The selective pressure exerted by these responses on rapidly mutating RNA viruses may drive the generation of immune escape mutations[1]. In chronic infections, epitope-specific T cell responses may acquire an exhausted phenotype after sustained antigen exposure, which is characterised by reduced responsiveness to further stimulation[2]. Pioneering work in the murine lymphocytic choriomeningitis virus (LCMV) model has mapped the molecular and phenotypic profiles of CD8+ T cells with acute resolving and chronic infections[3–5], revealing progenitors of exhausted T cells, defined by the expression of transcription factors, TOX and TCF1, which arise in the acute-phase of infection and sustain terminally exhausted subsets over the long-term[6–9]. A similar subset of progenitor has been identified within human memory populations[10], although how these cells arise and from which antigen specificity remains poorly defined.

[1]School of Medical Sciences, University of New South Wales, Sydney, NSW, Australia. [2]Kirby Institute for Infection and Immunity, University of New South Wales, Sydney, NSW, Australia. [3]Microsoft, Doha, Qatar. [4]Garvan Institute for Medical Research, Sydney, NSW, Australia. [5]Peter Doherty Institute, University of Melbourne, Melbourne, VIC, Australia. [6]School of Human Sciences, University of Western Australia, Perth, WA, Australia. [7]Institute for Immunology and Infectious Diseases, Murdoch University, Perth, WA, Australia. [8]Division of Infectious Diseases, Vanderbilt University Medical Center, Nashville, TN, USA. [9]These authors contributed equally: Curtis Cai, Jerome Samir. ✉e-mail: luciani@unsw.edu.au

Studies of human T cell responses to chronic viral infections are limited by the scarcity of samples from the acute-phase as many infections progress asymptomatically. Furthermore, the diversity of epitope targets, host HLA heterogeneity, and viral evolution require comprehensive analysis of immunological and virological variables to successfully identify individual-specific responses and their molecular and phenotypic profiles. Hepatitis C virus (HCV) infection is an excellent immunological model for understanding the development of exhaustion because both spontaneous clearance and chronic infections are naturally observed. The current model of the function of cytotoxic CD8[+] T cells in contributing to HCV disease outcome assumes that along with host factors, T cell's contributions to disease resolution are mediated by strong responses to a broad range of epitopes[11–13]. Recent studies on antigen removal via antiviral therapy against HCV or following immune escape described a scar on the recovered memory CD8[+] T cell responses, suggesting that exhaustion may not be fully reversible[8,9]. While human studies on the late-phase of HCV infection greatly advanced our understanding of T cell exhaustion, how the early viral dynamics and magnitude of responses determine immune escape and early differentiation from progenitor cells remains incompletely characterised.

In this study, we analyse longitudinal blood samples from a unique cohort of patients with primary HCV infections[14,15] within weeks of transmission and through their peak of viremia until determination of clearance or chronic infection. We apply deep sequencing of viral populations, functional, and single-cell multi-omics to characterise viral dynamics and HCV-specific CD8[+] T cell responses. In this analysis, the magnitude of IFN-γ responses during acute-phase of chronic infection is associated with early differentiation into exhaustion and positively correlates with clonal expansion and rate of immune escape. We identify a progenitor state of exhausted cells, irrespective of disease outcome, which persists after viral clearance. Our data contribute to understanding when and how exhausted T cell populations are formed and provide a revised model on the activity of CD8[+] T cells in determining the outcome of infection.

## Results

### High magnitude of IFN-γ response is associated with rapid viral immune escape

We applied a comprehensive combination of assays to study viral evolution and the molecular, phenotypic and functional characteristics of HCV-specific CD8[+] T cell responses from longitudinal samples during the acute-phase of primary HCV infection and before outcome is designated (Fig. 1a). Circulating viral genomes were deep sequenced longitudinally in both chronic progressors (CHs) and clearers (CLs) from the first viraemic samples to identify the transmitted virus genomes and the evolving viral variants that are generated during the infection (Supplementary Table 1). These genomes were utilised to identify Human Leucocyte Antigen class I (HLA-I) epitopes from the transmitted founder viruses and the mutated variants[16,17]. These epitopes were then experimentally tested using IFN-γ ELISpot (Fig. 1b, Supplementary Table 2). With this approach, we identified 50 epitopes across 14 individuals (Supplementary Table 3). Notably, 20 of these epitopes acquired amino acid mutations during the acute-phase of the infection, and all eventually became the dominant variant in the viral populations (defined as >75% of the total virus). These escape epitopes were detected exclusively in CHs (Fig. 1b). During the early-stage (defined as early phase, i.e., the first 120 days post-infection (DPI)), there was no significant difference in the levels of viral load (Fig. 1c, Supplementary Fig. 1a) and viral diversity across the full genome of HCV (Fig. 1d, Supplementary Fig. 1b), suggesting a similar viral dynamic across disease outcomes within the timeframe when acute viremia is successfully controlled in CLs and rebounds in CHs[17].

We next studied the kinetics of the epitope-specific immune responses and found that the magnitude of IFN-γ responses from CHs were higher than those identified in CLs between 90 and 120 DPI (Fig. 1e). The magnitude of the responses in CHs declined significantly over time, with some being associated with higher variability, and this decline was associated with epitopes undergoing viral mutations, while responses targeting conserved epitopes remained sustained (Fig. 1f). The IFN-γ SFU values were heterogeneously distributed across epitopes (Fig. 1g) and with no evident separation by disease outcome or HCV genotype (Supplementary Fig. 1c, d). Notably, for ten epitope-specific responses identified in three individuals, we measured IFN-γ responses against the mutated epitope sequences, which demonstrated reduced T cell response against escape variants (Supplementary Fig. 1e, Supplementary data 1). These results show that the IFN-γ producing cells occur in higher number in the acute-phase of CHs when compared to CLs, and rapidly decline over time in the presence of immune escape, while in CLs these responses remain sustained.

To investigate the relationship between the timing of immune escape and the magnitude of IFN-γ response, we first estimated the rate at which viral epitopes undergo immune escape by fitting the frequency of amino acid mutations that were identified within epitopes to a mathematical model of virus evolution with T cell killing (see Supplementary note)[18]. Estimated rates ranged from 0.03 to 0.51 day$^{-1}$ (Supplementary Table 4), which implied a broad time scale of 28–460 days for the observed escape variants to comprise over half of the total viral population. Notably, the peak of IFN-γ responses measured in this study preceded the estimated time at which escape variants became dominant (Fig. 1h), which provides strong evidence that the early high magnitude response contributes to rapid immune escape. We further observed a significant positive association via linear regression between the rate of escape and the peak-value of IFN-γ production (identified within the first 120 DPI), thus confirming the association between the magnitude of the IFN-γ response and the early onset of immune escape (Fig. 1i).

### Subsets of progenitors and exhaustion during acute-phase infection

We next investigated the phenotype of HCV-specific responses over the course of the infection, utilising dextramers to longitudinally study the phenotype of 20 epitope-specific CD8[+] T cell responses that were identified in five CLs and seven CHs (Supplementary Table 5). There was a strong positive correlation between the size of HCV-specific populations detected by IFN-γ ELISpot and dextramer staining (Supplementary Fig. 2a). Notably, larger populations of cells (per million PBMC) were detected by dextramer staining when compared to the number of cells that were associated with IFN-γ response, which proves that only a subset of cells were functionally responsive.

To investigate the phenotype of epitope-specific T cells, we employed high dimensional flow-cytometry and utilised two panels of antibodies to study the differentiation, activation, exhaustion and transcription factors related to T cell activity (see Supplementary Note 1, Supplementary Fig. 2b). We utilised co-expression of CD127 and PD-1 proteins to investigate the canonical subsets of HCV-specific T cell phenotypes, and identified five distinct subsets including exhausted PD-1$^{high}$CD127$^{low}$ ($T_{EX}$), memory PD-1$^{low}$CD127$^{high}$ ($T_{MEM}$), and effector PD-1$^{low}$CD127$^{low}$ ($T_{EFF}$) subsets (Fig. 2a-b). We observed two subsets of progenitor of exhausted cells, the first already previously reported as memory-like ($T_{ML}$)[8,19] with PD-1$^{high}$CD127$^{high}$, and a second with an intermediate level of PD-1 and low expression of CD127, herein referred to as ($T_{PINT}$), which has been previously reported in mouse models as an early progenitor of exhausted cells with responsiveness to anti-PD-1 treatment[20,21]. These two subsets were found in both disease outcomes, with $T_{PINT}$ in higher proportions than $T_{ML}$ (Fig. 2c). Within the first 120 DPI, higher proportions of $T_{ML}$ were observed within CL, while CH had increased proportions of $T_{EFF}$ subsets (Fig. 2d). Variability between epitopes was evident across all subsets, notably the HYP HLA-A*03:01 epitope in CH-3256 expressed high levels of $T_{EFF}$ and CD160$^+$

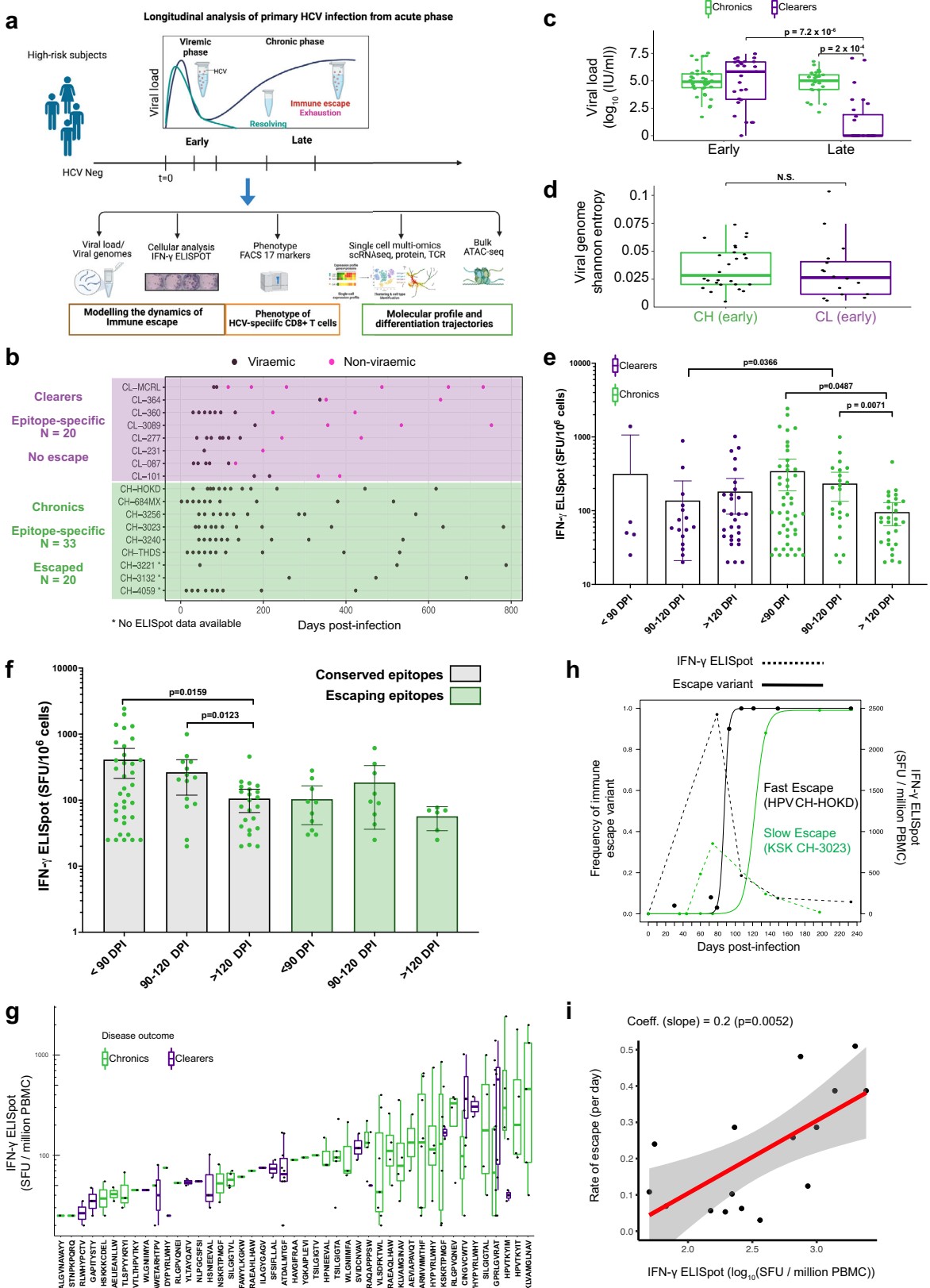

phenotypes (Supplementary Fig. 3a). Analysis of exhaustion and activation markers as well as of transcription factors in the early-stage showed increased proportions of T-bet$^+$ and of T-bet$^+$Eomes$^-$ but lower proportions of $T_{MEM}$ $T_{CM}$ and $T_{ML}$ subsets in CHs (Fig. 2d, Supplementary Fig. 3b). Notably, in this early-stage, CLs and CHs did not differ significantly in the expression of other exhausted associated markers,

such as CD160$^+$ and 2B4$^+$ subsets (Supplementary Fig. 3b). Comparison between the viraemic samples of CHs and CLs confirmed these findings (Supplementary Fig. 3c), thus confirming the increased levels of T-bet in CHs.

We performed an analysis of variance to investigate the effect of epitope specificity and HCV genotype on the distribution of T cell

**Fig. 1 | Comprehensive functional and multi-omics analyses show early and functional HCV-specific CD8⁺ T cells associated with viral immune escape.**
**a** Experimental design and details on the longitudinal study cohort utilised in this study (created with biorender.com). **b** Swimming plot outlining individuals ($N = 17$) followed longitudinally from early-phase of infection and in both disease outcomes. Sample viremia and the total number of epitopes detected are shown. **c** Comparison of viral loads by phase of infection (early: ≤120 DPI; late: >120 DPI) and disease outcome (two-sided Wilcoxon signed-rank test). Data are presented as box plots showing the median and 75% quantile. $N = 147$ biologically independent measures were used. Group comparison tests were performed utilising. **d** Comparison of viral genome diversity measured as Shannon entropy across disease outcomes in the early (≤120 DPI) phase of infection, group comparison tests were performed utilising two-sided Wilcoxon signed-rank test. Box plots show median and 75% quantile ($N = 56$ sample time points). **e** Comparison of IFN-γ responses according to the sample timepoint and disease outcome (two-sided

Mann-Whitney test). $N = 148$ independent measures from each unique individual, epitope, and timepoint combination. Shown are the median and 95% confidence interval. **f** Comparison of IFN-γ responses in chronic progressors ($N = 7$ individuals, $N = 99$ biologically independent measures) divided by epitopes that undergo immune escape or remain conserved during the infection. Shown are the median and 95% confidence interval. Comparisons were performed using two-sided Mann-Whitney test. **g** IFN-γ ELISpot values for samples from each epitope-specific immune response. Data are presented with box plots showing as median and 75%, $n = 154$ biologically independent measures were used. **h** Estimate of the rate of immune escape (continuous line showing model fit) and kinetics of IFN-γ response (SFU values), connected with dashed lines. **i** Correlation between the estimated rates of immune escape for each identified epitope and the corresponding maximum IFN-γ ELISpot responses measured within the first 120 DPI. All measures were obtained from chronic progressors. Data were fitted with a linear regression model, and $R^2$ and $p$ value are reported.

---

subsets. Both HCV genotype and epitope specificity contributed to the variability observed in each disease outcome (Supplementary Fig. 4). These effects were confirmed by performing a multi-variate regression of T cell subtypes over time, accounting for epitope and HCV genotype as covariates (Supplementary data 2). Genotype- and epitope-specific effects were found in $T_{EFF}$, $T_{PINT}$, and $T_{MEM}$ subsets and in both disease outcomes (Supplementary Fig. 4). Notably, the genotype 1a was associated with higher proportions of $T_{EFF}$ and lower $T_{PINT}$ in CHs, while in CLs, $T_{PINT}$ were higher in genotype 1a compared to genotype 3a.

In order to quantify the kinetics of T cell subsets over the course of the infection, we utilised longitudinal data for each epitope-specific response and fitted a linear regression model to the experimentally measured proportions of T cell subsets (Fig. 2e). This analysis showed that a rapid decline of $T_{EX}$, T-bet⁺, and CD38⁺ subsets in CH, while $T_{EFF}$ increased over the course of infection. In contrast, no significant declines were observed in CLs, with the exception of the PD-1⁺CD38⁺ subset, which showed a temporal decline irrespective of disease outcome (Supplementary Fig. 5).

The investigation of the relationships between phenotypes of HCV-specific T cells and the number of IFN-γ producing T cells showed that CHs had a positive association between the magnitude of response and the proportion of $T_{EX}$, $T_{ML}$, and activated CD38⁺ cells, while the $T_{EFF}$ subset decreased with the magnitude (Fig. 2f). In contrast, the proportion of $T_{PINT}$ was positively associated with the magnitude of response in CLs, and a similar trend was found for subsets expressing Eomes⁺, and exhaustion associated 2B4⁺, and CD160⁺ (Fig. 2f, Supplementary Fig. 5). Altogether, these results demonstrate that HCV-specific populations in CHs are characterised by a larger expansion of IFN-γ producing cells, consistent with an early onset of exhausted-like cells, expressing T-bet, but no significant differences in PD-1 and exhausted-associated markers CD160 and 2B4 compared to CLs.

## Single-cell multi-omics indicate molecular and phenotypic heterogeneity of HCV-specific CD8⁺ T cells

The identification of distinct subsets of progenitor of exhausted cells, $T_{PINT}$ and $T_{ML}$ within HCV-specific populations of CD8⁺ T cells demonstrated phenotypic heterogeneity, which we further explored with single-cell RNA-sequencing (scRNA-seq). To quantify the gene expression profile in phenotypically distinct T cell subsets and to reveal clonally expanded populations, we combined scRNA-seq with protein expression captured by index sorting during flow cytometry (see Supplementary note), and combined these with clonal data of full-length T cell receptor sequences reconstructed with VDJPuzzle[22] (Fig. 3a). In total, we analysed 1603 single cells from nine individuals (seven CHs and two CLs) (Supplementary data 3).

To quantify molecular heterogeneity, the scRNA-seq data was normalised and integrated together to account for technical noise (see Supplementary note). Dimensionality reduction (UMAP) and clustering of scRNA-seq data revealed a heterogeneous distribution of cells

forming ten clusters, which were characterised by a clear separation based on disease outcome, stage of infection (early vs late) and magnitude of IFN-γ response (Fig. 3b). T cells targeting epitopes undergoing immune escape were also located within clusters comprising early cells from CHs and high magnitude responses (Supplementary Fig. 6a). The distribution of T cell phenotypes within each cluster revealed an increased proportion of terminally differentiated $T_{EX}$ cells in clusters 4 and 5 which were predominantly found from samples in early-stage of infection and with high IFN-γ response (Fig. 3c, Supplementary Fig. 6a). In contrast, clusters 0, 2, and 8 comprised of cells from late-stage, were mostly $T_{MEM}$, $T_{ML}$, and $T_{PINT}$. Epitope specificity did not explain the observed cluster distribution (Supplementary Fig. 6b).

Differentially expressed genes between clusters revealed additional heterogeneity, with clusters from early time points (clusters 3, 4, 5, and 9) characterised by activating, cytotoxic and exhaustion gene signatures, while clusters comprising late-stage cells expressed a memory signature (Supplementary Fig. 6d, Supplementary data 4). Cells in clusters 0 and 2 expressed memory markers (e.g., *CCR7* and *SELL*) and transcription factors *TCF7* and *FOS* but differed in their expression of *JUNB, IL7R, CXCR4* and *NR4A2* (Supplementary Fig. 6d). Cells in cluster 5, with the largest proportion of $T_{EX}$, carried a cytotoxic gene signature (e.g., *IFNG, GZMB, PRF1*) as well as NK-like markers (e.g., *NKG7, KLRC2, KIR2DL1, KLRD1*, and *KLRB1*) and exhaustion (*PDCD1, CTLA-4, HAVCR2* and *TOX*). Early clusters (3, 4, and 5) expressed increased levels of activation (*MKI67, CD38* and *HLA-DR*) and exhaustion markers, but reduced expression of transcription factors *FOS, JUN, LEF1*, thus confirming increased differentiation. The smaller clusters, 6, 7 and 8, despite having similar disease stage and phenotype distributions (Fig. 3b, c), differed in their expression of *TCF7, LTB* (encoding tumour necrosis factor TNF-C) and killing like receptors *NKTR, KLRB1*, and *NKG7* (Supplementary Fig. 6d). Cluster 8 had increased expression of transcription factors *FOS, JUNB* and *BACH2* which are known to be involved in blocking T cell activation and differentiation[23–26]. GSEA confirmed these findings (Supplementary Fig. 6e), specifically revealing cytotoxic CD8⁺ and NK-like enrichment in clusters 3 and 4, along with an enrichment of IFN-γ response and metabolic activities, such as fatty acid metabolism, glycolysis and oxidative phosphorylation (Supplementary data 5, 6) in cluster 5 as well. In contrast, late-stage clusters revealed a progenitor of exhausted signature, effector memory and were negatively enriched for cytotoxic and NK-like signatures.

As T cell subsets were distributed across all clusters, we reasoned that additional heterogeneity could be identified between the five T cell subsets. Firstly, we confirmed consistency between gene and protein expression, by comparing mean fluorescence intensity (MFI) index sorting values with the corresponding gene expression, for all the available pairs, including CD127, PD-1, CD38, and CD95 (Fig. 3d). We then characterised the gene profiles by calculating differentially expressed genes between each subset and found that $T_{PINT}$ cells had an intermediate transcriptomic profile between $T_{EX}$ and $T_{ML}$ (Fig. 3e, f).

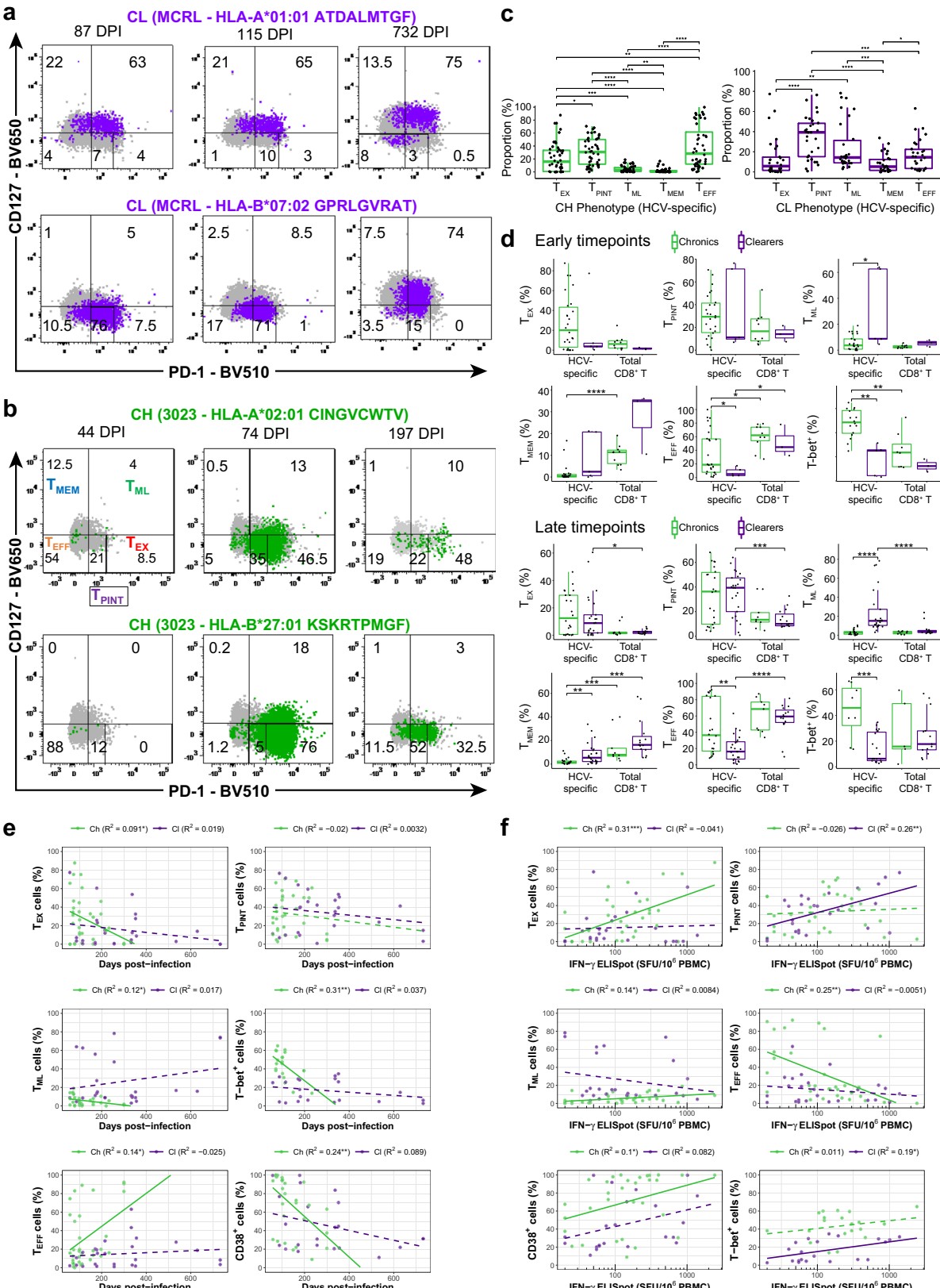

Despite $T_{EX}$ sharing the largest number of differentially expressed genes with $T_{ML}$, the largest proportion of the up-regulated genes were shared with $T_{PINT}$ (Fig. 3e). When compared to $T_{EX}$ cells, both $T_{PINT}$ and $T_{ML}$ were not enriched for exhaustion genes (*TOX, TIGIT, LAG3,* and *PDCD1*), and expressed higher levels of *IL7R, TCF7, FOS* (Fig. 3f, Supplementary data 4). GSEA analysis showed that $T_{PINT}$ had lower

enrichment scores than $T_{EX}$ for gene signatures associated with exhaustion signature and had higher cytotoxicity scores when compared to $T_{ML}$ (Supplementary Fig. 7a, Supplementary data 5, 6). Transcription factors known to be associated with T cell differentiation and function confirmed the intermediate differentiation state of $T_{PINT}$ and $T_{ML}$ subsets between $T_{MEM}$ and $T_{EX}$ (Supplementary Fig. 7b).

**Fig. 2 | Subsets of progenitors of exhausted T cells during the acute-phase of infection.** Representative flow cytometry gating plots of dextramer-positive CD8⁺ T cell populations analysed longitudinally targeting autologous epitopes identified from circulating viral populations, in clearers (CL) (**a**) and chronic progressors (CH) (**b**). Each plot shows dextramer positive cells (coloured) and the frequency of phenotypic subsets based: $T_{EX}$ (PD-1$^{high}$CD127$^{low}$), $T_{PINT}$ (PD-1$^{int}$CD127$^{low}$), $T_{ML}$ (PD-1$^{high}$CD127$^{high}$), $T_{MEM}$ (PD-1$^{low}$CD127$^{high}$), and $T_{EFF}$ (CD127$^{low}$PD-1$^{low}$). DPI: days post-infection. Shown are cells gated from a lymphocyte/singlet/live population that were CD19 negative, CD3, and CD8 positive. **c** Group comparison of phenotypic subsets of HCV-specific populations segregated by disease outcome. Individual points represent unique individual, timepoint and epitope specificity. Data are presented as median and 75% quantile. $N = 116$ biologically independent blood samples and 10 epitope specificities. **d** Comparisons of the proportion of cells with positive expression of protein markers measured by flow cytometry between dextramer positive and total CD8⁺ T cells in both disease outcomes, split into early (≤120 DPI) and late (>120 DPI) time points. Individual points represent populations from each individual's sample time points, and from each epitope specificity, $N = 44$ biologically independent samples were used. Data are presented as median and 75% quantile. Pairwise group comparisons were performed utilising a two-sided Wilcoxon signed-rank test. **e** Plots showing the proportion of dextramer-positive CD8⁺ T cell populations with positive expression of phenotypic markers (measured by flow cytometry) over the course of the infections (DPI values). Lines represent linear regression, and adjusted coefficients and $p$ values are shown in the legend of each plot. Continuous lines (dashed) represent significant (not significant) linear fits (p-value <0.05). **f** Similar to **e**, plots of the proportion of dextramer positive CD8⁺ T cell populations with positive expression of phenotypic markers (measured by flow cytometry) and IFN-γ ELISpot values (number of SFU per million PBMC) on a log₁₀-scale.

We next compared the gene signatures of progenitors of exhausted cells $T_{PINT}$ and $T_{ML}$ with those previously identified in the mouse model of LCMV infection[27,28], as well as with T cell exhaustion profiles from human tumour infiltrating lymphocytes[29]. We annotated our scRNA-seq data with the reference data set on progenitor cells from Miller et al.[28] (Supplementary Fig. 8), and found that $T_{PINT}$ and $T_{ML}$ were mostly correlated with the subset of progenitor of exhausted cells, while $T_{EX}$ revealed a correlation with both progenitor and proliferating subsets. Notably, $T_{PINT}$ and $T_{ML}$ retained higher levels of expression of AP-1 transcription factor when compared to $T_{EX}$ (Supplementary Fig. 7b). Comparison with human exhaustion signatures derived from TILs in liver cancer revealed that $T_{EX}$ were closely related to the original subset of exhausted cells, while $T_{PINT}$ and $T_{ML}$ overlapped with the gene signatures of T cells identified in the original study as effector as well as CD8⁺LEF1⁺ cells identified as resting cells (naïve or memory)[29]. This analysis revealed that $T_{PINT}$ and $T_{ML}$ are both consistent with known profiles of progenitor of exhausted cells, however, these differ from the subsets identified in the mouse model of LCMV, which are not terminally differentiated exhausted cells.

We next stratified cells into high and low IFN-γ responses with a threshold of 211 Spot forming units (SFU)/million PBMC - the 85th percentile of recorded values. High IFN-γ producing cells expressed exhausted-like (*PDCD1, LAG3, HAVCR2, CD160*), transcription factors *TBX21*, activation markers (*HLA-DR* and *CD38*), TCR signalling (*LCK*), and functional (*GZMA, GZMB, NKG7*, and *IFNG*) genes (Supplementary data 4), while low IFN-γ responses expressed memory-like profiles (*IL7R* and *SELL*) and transcription factors *TCF7, FOS, NR4A2*. Correlation analysis between gene expression and magnitude of IFN-γ response showed that genes associated with cytotoxicity, such as *NKG7, IFNG, GZMA*, and *TBX21*, increased with the magnitude of response, while an opposite trend was observed for transcription factors *FOS* and *JUNB*, as well as *IL7R* (Fig. 3g).

Comparisons across the early- and late-stages of infection showed that cells from CHs had an early expression of genes associated with exhaustion (*PDCD1, ENTPD1*), and cytotoxicity, including NK-like genes *NKG7, GNLY, KLRC2*, and *KLRC3* (encoding for NKG2C and NKG2E, respectively) (Fig. 3h). These cells also expressed higher levels of *TBX21, TOX*, and activation markers *CD38* and *HLA-DR* (Supplementary data 4). In contrast, CLs in the early-stage had a higher expression of transcription factors *FOS* and *JUN*, and memory-like associated genes (*IL7R, TCF7, CCR7*), which also persisted in the late-stage of infection where cells had higher expression of transcription factor *BACH2* and of *S1PR1* (encoding sphingosine-1-phosphate receptor-1), which is known to promote T cell retention in non-lymphoid tissues[30]. GSEA analysis revealed an enrichment of oxidative phosphorylation, proliferation, cytotoxicity, NK-like and exhaustion gene signatures in early-stage CHs, while cells in CLs showed a sustained enrichment of progenitor of exhaustion, and memory profiles (Supplementary Fig. 7c, Supplementary data 5, 6).

By reconstructing full-length TCR sequences from scRNA-seq data[10], we observed increased clonal expansion in clusters formed by cells from early-stage and associated with high IFN- response (Supplementary Fig. 6a). The distribution of clone size was highly variable, ranging from highly diverse to nearly monoclonal populations, which were found in HLA-B*07:01 GPR (CL-MCRL) and HLA-A*03:01 HYP (CH-3256) epitopes, respectively (Fig. 3i, Supplementary Table 6). Over half of the cells formed expanded clones, and 63 (46%) clones persisted through both the early- and late-stages of infection, displaying a mix of phenotypes (Fig. 3j). The early-stage was characterised by expanded clones mostly with a $T_{EX}$ or $T_{PINT}$ phenotype, while clonal expansion during the late-stage was mostly distributed between $T_{ML}$ and $T_{PINT}$. We measured the level of clonal diversity utilising Shannon Evenness scores (SEv), and found reduced diversity (i.e., larger clonal expansion), in responses with a high magnitude of IFN-γ, which was also supported by linear regression analysis ($R^2 = 0.12$, $p = 0.046$) (Fig. 3k, l). These results demonstrate that clonal expansion occurred mostly within high IFN-γ responses and predominantly within progenitor of exhausted cells, and notably in the early-stage, characterised by a $T_{EX}$ phenotype.

Altogether, single-cell multi-omics analysis demonstrated that the acute-phase of HCV infection is characterised by heterogeneous populations of progenitor of exhausted cells with an activated and cytotoxic phenotype and an intermediate differentiation state between $T_{ML}$ and $T_{EX}$.

## Distinct evolutionary trajectories of progenitor of exhausted cells in acute-phase of infection explain functional heterogeneity in both disease outcomes

To further investigate the molecular differences observed in the UMAP and clustering analyses between outcomes and between the stages of infection, we reasoned that distinct differentiation trajectories underline the dynamics of HCV-specific T cell responses in each disease outcome. Firstly, we performed Slingshot trajectory analysis using the UMAP clusters (as in Fig. 3) to generate pseudotime ordering of cells (Supplementary Fig. 6f). We selected cluster 5 as a root, as it was comprised of cells from early sample time points, and identified four trajectories. Two trajectories terminated in clusters formed by late-stage cells (7 and 8), and two remained within early-stage clusters, terminating in clusters 3 and 5, respectively. The two trajectories ending in clusters 7 and 8 were consistent with a differentiation process from an early activated state into an effector (cluster 7) or memory state (cluster 8), while the other two trajectories were consistent with differentiation into activated and proliferating cells, in line with the phenotypic distribution of early-stage clusters. This analysis revealed that molecular heterogeneity of the HCV T cell responses is consistent with more than one differentiation process and that functional state and stage of infection are important factors associated with each differentiation trajectory.

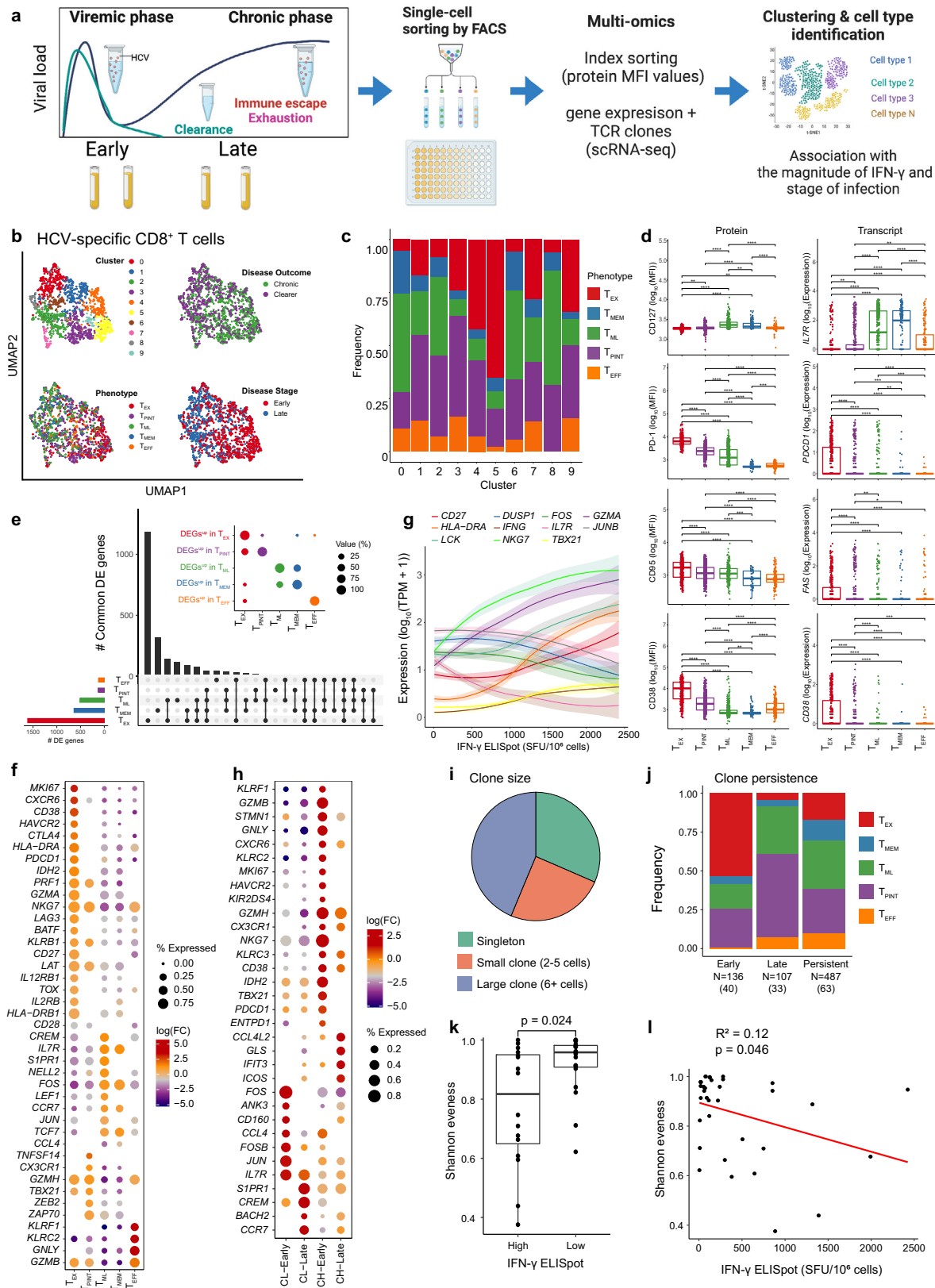

To take advantage of the longitudinal sampling time points, we utilised a bioinformatic approach to infer cellular state by estimating differentiation trajectories and cluster connectivity informed by prior information on the root and terminal state which were set a priori from the longitudinal data (Fig. 4a). Similar to previous T cell studies with longitudinal data[31–33], we employed a partition-based graph abstraction (PAGA) (Fig. 4b) which allows inference of the connectivity between clusters. PAGA dimensionality reduction was consistent with the UMAP analysis, revealing a close transcriptional signature between cells according to the stage of infection (Fig. 4c), the magnitude of IFN-γ production, as well the rate of immune escape (Fig. 4d). Cells from responses targeting epitopes with a

**Fig. 3 | Single-cell multi-omics show molecular and phenotypic heterogeneity of HCV-specific CD8⁺ T cells. a** Experimental design with single-cell multi-omics analysis (created with biorender.com). **b** Dimensionality reduction (UMAP) and clustering of scRNA-seq data. **c** Distribution of T cell phenotypes in each UMAP cluster. Single-cell phenotypes are obtained via index sorting protein expression data. **d** Comparison of protein (left) and corresponding genes (right) expression levels between cells grouped by their phenotypes ($N = 1603$ cells from 26 biologically independent sample time points across 7 epitope specificities). Pairwise group comparisons were performed with two-sided Wilcoxon Rank Sum Tests (*$p < 0.05$, **$p < 0.01$, ***$p < 0.001$, ****$p < 0.0001$). Data are presented as median and 75% quantile. **e** Upset plot showing the number of commonly differentially expressed genes between T cell phenotypes. Insert shows the proportion of genes upregulated between indicated T cell phenotypes. **f** Selected genes identified from differential expression analysis between cells grouped by phenotype (pairwise comparisons with a two-sided hurdle-model (MAST[72]). **g** Loess curve fit of selected genes as a function of the magnitude of IFN-γ response, shaded regions represent 95% confidence intervals. **h** Selected genes identified from differential expression analysis between cells grouped by disease outcome and phase of infection (early: ≤120 DPI, late >120 DPI). CL: Clearers, CH: Chronic progressors. **i** Distribution of the clone size and **j** in each T cell phenotypic subset. Shown are clones found only in the early (≤120 DPI), late (>120 DPI), or both phases (persistent) of infection. *N* represents cell numbers, and in brackets is the unique number of clones. **k** Group comparison of T cell receptor (TCR) diversity measured by Shannon evenness between high and low magnitude of IFN-γ responses ($N = 22$ sample timepoints, 9 epitope specificities). Statistical comparison is performed with a two-sided Wilcoxon Rank Sum Test. Data are presented as median and 75% quantile. **l** Linear regression (drawn in red) of the Shannon evenness as a function of the magnitude of IFN-γ.

faster rate of escape (clusters 0, 7, and 8) featured larger clonal expansion when compared to other clusters (Fig. 4d) and could not be attributed to any single exhaustion phenotype (Fig. 4e). Cluster genes revealed additional heterogeneity in early cells, for instance, clusters 4, 6 formed by early-stage cells targeting conserved or slow-escape epitopes revealed cytotoxicity genes (*GNLY*, *NKG7*, *GZMB*), as well as expression of *FOS* and *JUN*. In contrast, clusters 0 and 8 had cells targeting fast-escape epitopes with higher levels of activation and proliferation markers (*CD38*, *HLA-DR*, *MKI67*), and reduced cytotoxicity but an increased level of exhaustion markers (*CTLA4*, *TOX*, *ENTPD1*, *EOMES*) (Supplementary Fig. 9b).

From the PAGA structure, three distinct trajectories were identified, CH-T1, CH-T2, and CH-T3, with roots in early-stage clusters, 4, 8, and 3, respectively (Fig. 4f, Supplementary Fig. 9c) and terminated in late-stage cells (clusters 2, 7, and 12, respectively, Fig. 4c). Diffusion pseudotime values were calculated to infer the order of cell states along the trajectories[34]. CH-T1 described the differentiation of cells associated with slow or no epitope escape and with decreasing IFN-γ magnitude (Fig. 4d). Along this trajectory, CD38 and CD95 proteins increased as well as memory genes *IL7R* and *TCF7*, AP-1 transcription factors *FOS* and *JUNB*, while there was a decline in cytotoxicity genes (e.g., *GZMA* and *NKG7*) and relatively stable levels of exhaustion markers (Fig. 4f). In contrast, CH-T2 comprised of cells from high-magnitude IFN-γ responses, which had larger clonal expansion and targeted viral epitopes undergoing faster immune escape. It was associated with a rapid decline in CD38, CD95, and PD-1 protein expression, as well as of exhaustion markers (e.g., *LAG3*, *TIGIT*, *PDCD1* and *HAVCR2*), and *IFNG* and *LCK* genes (Fig. 4f), suggesting early activation and cytotoxicity. CH-T3 comprised cells associated with low magnitude IFN-γ responses, which were less clonally expanded, and targeted epitopes undergoing slower or no immune escape. This trajectory revealed relatively stable expression of CD38, PD-1 and CD95, as well as of *IL7R*, *FOS* and *JUNB* genes (Fig. 4f).

These trends were confirmed via a differential gene expression analysis between the three trajectories (Fig. 4g). Notably, CH-T1 was enriched for NK-like cytotoxicity in the early-stage (*GNLY*, *KLRC2*, *NKG7*, *KLRK1*, and *CD160*), along with increased expression of *GZMA* and *IL2RG*, while exhaustion markers were moderately expressed. GSEA confirmed the enrichment of cytotoxic, NK-like signatures as well as oxidative phosphorylation in the early-stage of CH-T1 (Fig. 4h), while CH-T2 was enriched for cell-cycle, proliferation and fatty acid metabolism in the early-stage (Supplementary data 5, 6). All three trajectories had a low enrichment of progenitor of exhaustion and expression of AP-1 transcription factors (*FOS*, *JUN*, *JUNB*) in the early-stage, confirming the differences in the magnitude of response in early-stage (Supplementary Fig. 3d). We quantified the velocity with which our T cell subsets changed over the pseudotime along the inferred trajectories (Supplementary Fig. 9c). CH-T1 and CH-T2 featured high velocities (growth rates) of $T_{EX}$ in early-stage, which then declined as $T_{PINT}$ and $T_{EFF}$ began to dominate in the late-stage. CH-T2,

associated with high magnitude IFN-γ responses and fast epitope escape, had the fastest decline in $T_{EX}$. In contrast, the low magnitude IFN-γ response in CH-T3 was dominated by $T_{PINT}$ cells in the early-stage, which was followed by a transition towards a $T_{EX}$ phenotype in the late-stage.

## Trajectory analysis in clearers revealed maintenance of $T_{PINT}$ and increase of $T_M$ and $T_{ML}$

Similarly, trajectory analysis was applied to single-cell multi-omics data from CLs (Supplementary Fig. 10), using 700 HCV-specific CD8⁺ T cells longitudinally sampled from viraemic early-stage to post-viral clearance. Consistent with the flow cytometric analysis in Fig. 2, most cells carried a memory or precursor phenotypes ($T_{MEM}$, $T_{ML}$, $T_{PINT}$) (Supplementary Fig. 10a). PAGA analysis revealed 13 clusters, with cells clustering according to early viraemic- or a late-stage post-clearance (Supplementary Fig. 10b, c). Cells also clustered by their epitope specificity, clearly separating HLA-B*0702 GPR - and ATD HLA-A*0101 -specific T cells. Cluster genes showed two distinct signatures for early cells which formed the two extreme edges of the PAGA structure (Supplementary Fig. 10d). Cells in clusters 2 and 6 showed increased expression of cytotoxic markers (e.g., *IFNG* and *GZMA)*, and NK-like markers (*NKG7* and *KLRC4-KLRK1*) when compared with the other early-stage clusters 9, 12, and 0. By combining connectivity between clusters and diffusion pseudotime, we discovered two trajectories, CL-T1 and CL-T2, with root clusters 6 and 9 (Supplementary Fig. 10f), each comprising of both epitope specificities. The differentiation trajectory in CL-T1 was characterised by large clones, early expression of CD38 and CD95, and an increase of CD127 protein expression, clearly revealing a transition from an early effector to a memory phenotype following viral clearance. These phenotypic kinetics were further corroborated by the decreasing expression of cytotoxic genes *IFNG*, *GZMA*, and *NKG7*, exhaustion markers, and transcription factors *FOS* and *JUN*. CL-T2, in contrast to the first trajectory, featured an initially higher, albeit decreasing, expression of PD-1 and KLRG1 proteins, as well as a reduced cytotoxicity signature and increasing expression of transcription factors *FOS* and *JUN*. The phenotypic growth rates (velocities along trajectories) confirmed the transition from the early, viraemic-phase with high proportions of $T_{ML}$ and the more cytotoxic and exhausted $T_{PINT}$, to an increase of $T_{MEM}$ and sustained $T_{ML}$ following viral clearance (Supplementary Fig. 10f). Differential gene expression analysis between the early- (including viraemic sample time points) and late- (post-clearance) stages confirmed an early cytotoxic active cell state in cells from CL-T1 as well as of exhaustion genes (*TOX*, *KLRG1*, and *EOMES*) when compared to early cells in CL-T2 (Supplementary Fig. 10g). Notably, CL-T1 revealed early expression of transcription factor *JUN* when compared to late-stage cells, which is in contrast to the lack of these factors in early-stage of CHs (Fig. 4).

In summary, trajectory analyses by disease outcome described the differentiation process through cell states that explain the molecular differences between immune responses with varying magnitudes of IFN-γ response and epitope recognition. The early-stage of the

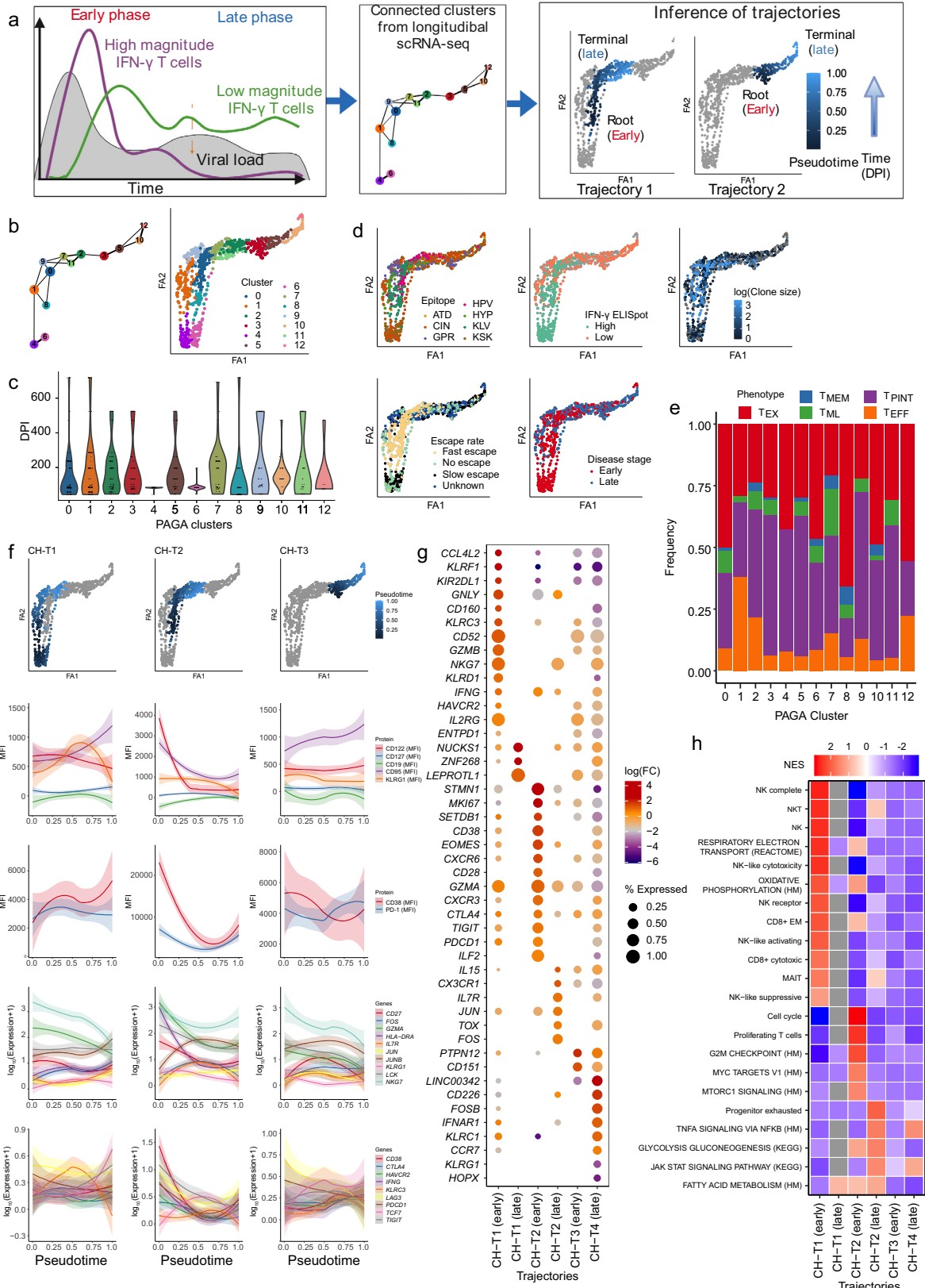

immune responses in CHs with fast immune escape rates are associated with rapid onset of $T_{EX}$, and lack of expression of *TCF7* and AP-1 transcription factors, which contrasted with trajectories from viral clearance or with slow escape rates where $T_{PINT}$ and $T_{ML}$ subsets dominate early-stage along with high expression of AP-1 transcription factors.

## Fate mapping implicates phenotypic plasticity within a clonally expanded population

The serendipitous discovery of a monoclonal response in the high IFN-γ GPR-specific T cells in one individual (CL-MCRL) permitted the investigation of intra-clonal fate mapping of a single lineage. We performed lineage tracing from 244 cells from 4 samples, two of which

**Fig. 4 | Distinct evolutionary trajectories identified from single-cell transcriptomics explain functional heterogeneity in chronic infections. a** Schematic of the method to identify differentiation trajectories from scRNA-seq data (created with biorender.com) utilising partition-based graph abstraction (PAGA), a priori information on longitudinal samples and IFN-γ ELISpot data. **b** PAGA graphs of single cells (n = 903) identifying 13 clusters and their probability of connection (thickness of lines). **c** Distribution of the estimated days post-infection (DPI) corresponding to the sample time point associated with each cell. **d** PAGA graphs (visualised using ForceAtlas2 layout algorithm (FA1, FA2)) coloured by epitope specificity, disease outcome, disease stage, magnitude of IFN-γ response, rate of escape (slow, fast, no escape), and clone size (log-transformed). **e** Distribution of T cell phenotypes across PAGA clusters (as per **b**). Single-cell phenotypes are obtained via index sorting protein expression data. **f** Trajectories derived from scRNA-seq data with coloured pseudotime values. Loess curves represent the changes along the trajectories of both single-cell protein and gene expression values. The 95% confidence intervals are denoted by shaded regions. **g** Dot plot of selected differentially expressed genes (derived from pairwise comparisons with a two-sided hurdle-model from MAST) between the early and late phases of each trajectory (p < 0.05, |log₂(FC)| ≥ 1, coloured). Dot size represents the proportion of cells with non-zero expression. Early ≤120 DPI, late >120 DPI. **h** Enriched pathways identified from GSEA using differentially expressed genes (as per (g)) between early and late phases within each trajectory. Adjusted p values <0.05 in at least one trajectory phase. NES: normalised enrichment score.

were early viraemic samples (day 88 and day 116) and two post-clearance (Fig. 5a). During viremia, cells were heterogeneously distributed across the phenotypes excluding $T_{EX}$, and following clearance a transition was observed towards memory phenotypes (Fig. 5b).

As this response was associated with a very high magnitude IFN-γ response in the early-stage, we reasoned that a subset of cells within this monoclonal population were responsible for the high magnitude response. We generated a PAGA structure from these cells, which revealed clusters segregated by stage of infection and phenotype (Fig. 5c, d). Ten clusters were identified, largely separated by disease stage and magnitude of IFN-γ response. Clusters 1 and 8 formed by a subset of early cells at 88 DPI were enriched for *IFNG* as well as *JUN*, while early cells forming cluster 2 expressed a higher level of *FOS* and lacked *IFNG* expression (Fig. 5e). Trajectory analysis revealed two distinct lineages M-T1 and M-T2 (Fig. 5f). M-T1, which lacked *IFNG* in the early-stage, had early elevated KLRG1, PD-1 and CD38 protein expression and was associated with high *JUNB* and increasing *FOS*, *JUN*, and *TCF7* expression. In contrast, M-T2 revealed lower levels of CD38 and PD-1 proteins and higher expression of *IFNG* than M-T1. These two trends were confirmed by higher growth rates for $T_{PINT}$ and $T_{ML}$ in M-T1 in early-stage, while M-T2 featured a decline of $T_{PINT}$ and increased levels of $T_{EFF}$ and $T_{ML}$, which is consistent with a more effector and cytotoxic response. The kinetics along trajectories were then confirmed by performing differential gene expression between the early and late-stages of each trajectory (Fig. 5g). M-T1 down-regulated *IFNG* in early-stage, while M-T2 expressed *JUN* in the early-stage, and then cytotoxic genes (e.g., *NKG7* and *GZMA*) in the late-stage.

In summary, this analysis traced the evolution of multiple T cell states within a single clone, demonstrating phenotypic plasticity and revealing two lineages undergoing rapid transition from progenitor of exhausted cells to memory subset, and only one responsible for IFN-γ secretion.

## Epigenetic heterogeneity associated with responses targeting conserved and escaping epitopes

We used bulk ATAC-seq on HCV-specific and total CD8⁺ T cells in the early-stage of infection and characterised the chromatin accessibility profiles of responses with high or low IFN-γ magnitude and against conserved or escaping epitopes. For this analysis, we utilised cells sampled at the earliest available time points (with matched scRNA-seq data) from four immune responses. Two responses were associated with high IFN-γ production and targeted epitopes undergoing fast escape (KLV in CH-THDS and HPV in CH-HOKD). The third response targeted a conserved epitope (CIN in CH-3023) and was associated with low IFN-γ production. The last response was the monoclonal population of T cells generating the highest IFN-γ production among CLs, targeting the conserved epitope GPR (in CL-MCRL). Principal component analysis (PCA) of chromatin accessibility profiles revealed that HCV-specific and control CD8⁺ subsets (effector memory, activated CD38⁺, or CD8⁺Dextramer⁻) clustered by individual of origin rather than specificity for viral epitope (Fig. 6a). As expected, the chromatin accessibility profile from the monoclonal response in CL-MCRL was separated from the other populations.

The comparison between the four HCV-specific responses against control subsets revealed 15 differentially opened regions, including the loss of opening in the loci of Latent-transforming growth factor beta-binding protein 1 (LTBP1), which is known to regulate TGFB release and is consistent with reduced immune-regulation in HCV-specific responses. Analysis of the most variable opening regions between the four HCV-specific populations identified significant loss of accessibility in GPR-specific T cells (CL-MCRL), notably in loci associated with exhaustion such as *ENTPD1* (encoding CD39), cytotoxicity (e.g., *CCL3, KLRC4-KLRK1*), and activation (*TNFRSF18*, encoding GITR) (Fig. 6b, Supplementary data 7), which is in line with reduced activation in T cells from individuals who cleared the virus as shown above. In contrast, GPR-specific T cells revealed an increased opening in *IFNA1* (encoding interferon alpha 1) and *HLA-DRB5*.

We next compared chromatin accessibility profiles to single-cell gene expression data (Fig. 6c). We identified opening profiles that were correlated with gene expression data. As expected, accessibility for transcription factors *FOS* and *JUN* were increased in GPR-specific T cells, while with loci encoding genes associated with TCR signalling (*LCK*), cytotoxicity (*IFNG, KLRC4*), and exhaustion (*ENTPD1, TOX, TIGIT*) were significantly more open in high IFN-γ, fast escape responses (Fig. 6c). Interestingly, the CIN-specific response (CH-3023) had similar levels of accessibility in exhaustion (e.g., *TOX, CTLA4, ENTPD1*), activation (*CD38*), and effector memory genes (*EOMES, CD27*) as the high IFN-γ responses in CH-THDS and CH-HOKD, despite the reduced gene and protein expression profiles corresponding to these loci (Fig. 6c). This result is consistent with a previous study, which performed ATAC-seq on cells targeting the same conserved epitope but in late-stage of chronic infection[35], which suggest that lack of immune escape is associated with increased chromatin opening in exhaustion associated genes. Finally, we compared the opening profiles from bulk ATAC-seq with the single-cell protein and gene expression data for CD127 and PD-1 (Fig. 6d, e). This analysis revealed a consistent pattern across the three datasets, with increased *IL7R* accessibility in the cells from CL-MCRL and increased opening, gene and protein expression of *CD38* in cells associated with high IFN-γ production (KLV in CH-THDS and HPV in CH-HOKD). Chromatin accessibility for *KLRG1* and *FAS* was negatively correlated with gene expression (Fig. 6d), while the *PDCD1* locus showed a similar opening profile across all samples. Notably, three of the four populations lacked the opening of the intragenic *PDCD1* cis-element associated with terminal exhaustion (Fig. 6e)[36], while one response (KLV-specific) revealed a low coverage peak. These results reinforced the conclusion that T cell responses analysed in this study are indeed dominated by precursors of terminally exhausted cells[35,37,38].

## Discussion

In this study, we analysed a unique collection of samples from individuals followed longitudinally from the onset of acute HCV infection

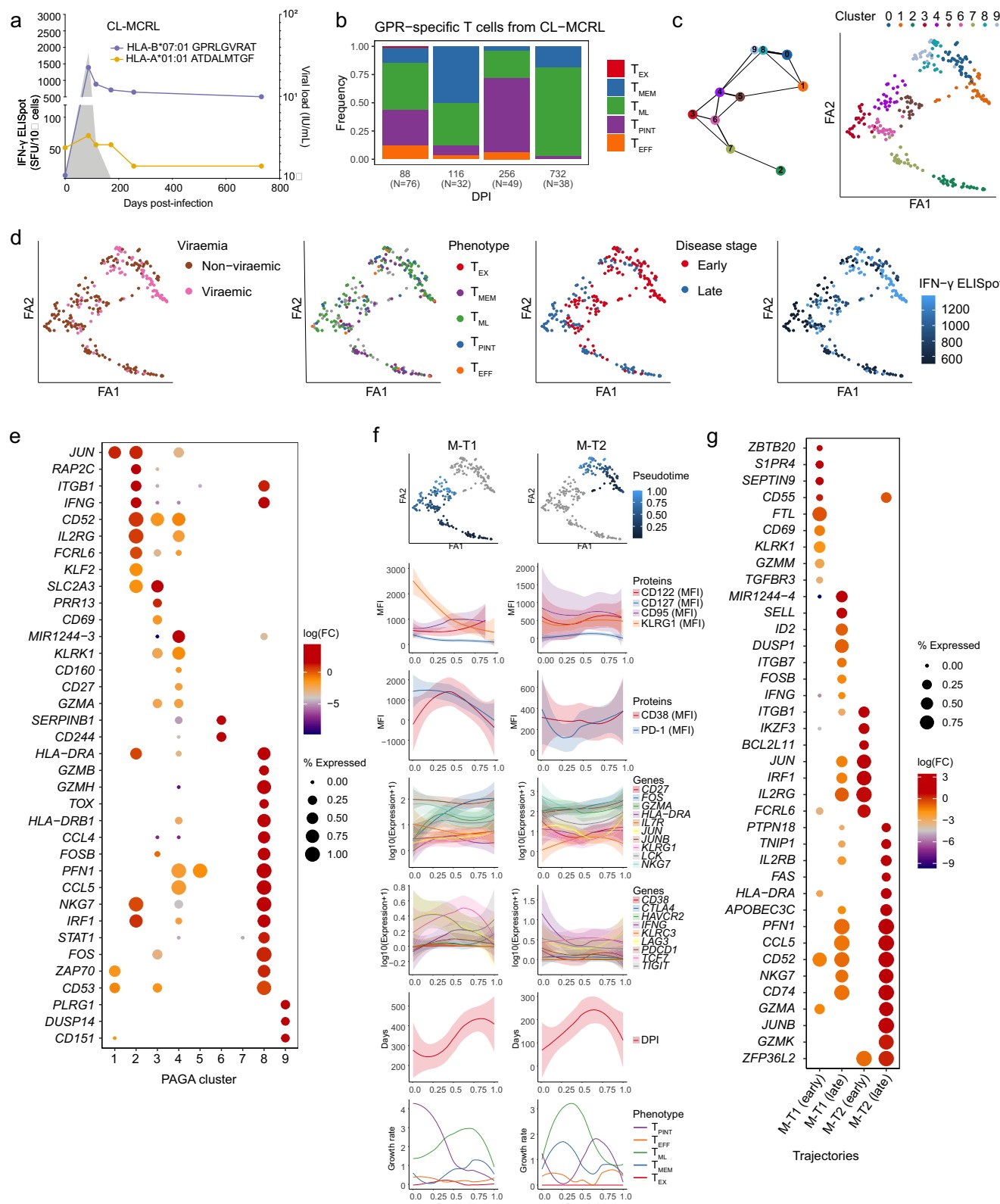

and implemented a multi-omics approach across virus and host to quantify the molecular, cellular and functional dynamics of epitope-specific CD8+ T cells. We combined functional and phenotypic single-cell profiling from both CHs and CLs to understand T cell differentiation in a model of disease with the potential for chronicity. In CLs, the early-stage was associated with $T_{ML}$ subsets, while chronic progressors revealed an early onset of $T_{EX}$, while the precursor subset $T_{PINT}$ was

found in both disease outcomes. As expected, irrespective of disease outcome, we found that the expression of genes associated with cytotoxicity and activation followed the magnitude of IFN-γ response, whilst memory markers and AP-1 transcription factors declined as the magnitude of the associated responses increased. Our data provide evidence that in the early-stage of chronic progression, elevated IFN-γ response and fast immune scape are associated with early onset of

**Fig. 5 | Fate mapping implicates phenotypic plasticity within a clonally expanded population. a** Kinetics of viral load (shaded areas, measured in international units (IU)) and IFN- production (Spot forming unit (SFU/Million PBMC)) for the GPR- and ATD-specific CD8[+] T cell responses identified in individual CL-MCRL. **b** Distribution of T cell phenotypes over time in GPR-specific cells identifying a monoclonal response (identical TCRαβ sequence). **c** PAGA graph representation and clustering of GPR-specific cells (N = 244) revealing ten connected clusters (line thickness represents the probability of connectivity). **d** PAGA graphs coloured by sample viremia, disease stage, T cell phenotype and IFN-γ ELISpot. **e** Dot plot of selected genes identified from differential expression analysis (derived from pairwise comparisons with a two-sided hurdle-model from MAST) on the PAGA

clusters. Only significant genes are shown (p < 0.05, |log₂(FC)| ≥ 0.3, coloured). Dot size represents the proportion of cells with non-zero expression. **f** Trajectories derived from scRNA-seq data with coloured pseudotime values. Loess curves represent the changes along the trajectories of both single-cell protein and gene expression values. The 95% confidence intervals are denoted by shaded regions. Growth rates are estimated from the calculated size of T cell phenotypes over pseudotime values for each trajectory. **g** Dot plot of selected differentially expressed genes (derived from pairwise comparisons with a two-sided hurdle-model from MAST) between the early and late phases of each trajectory (p < 0.05, | log₂(FC)| ≥ 0.3). Dall size represents the proportion of cells with non-zero expression. Early ≤120 DPI, late >120 DPI.

exhaustion and cytotoxic signatures, as well as with a higher metabolic activity of the cells, which contrasted with the reduced activation and function of cells leading to low magnitude responses, irrespective of disease outcome.

This study provides compelling evidence against the current model of a broad and high magnitude response predictive of viral clearance. Instead, optimal and sustained T cell responses may be best achieved with moderate activation which avoids immune-driven selection of escape variants and early onset of exhaustion. As previously shown, we have demonstrated that a genetic bottleneck in viral populations occurs at around three months post-infection, where the transmitted founder viruses are replaced with novel variants that dominate the populations and carry amino acid changes across the viral genome, including within HLA-I restricted epitopes[16,17]. These previous findings from our group support the conclusion that the magnitude of CD8[+] T cell responses influences viral evolution in the acute-phase of primary HCV infection and the onset of escape variants.

The comprehensive data generated from this and our previous findings provide evidence for a revised model of CD8[+] T cell differentiation in the early-stage of viral infection. In this model, the main assumption is that timing and magnitude of clonal expansion is a function of stochasticity in antigen stimulus[39], TCR affinity for antigen[40,41], and other factors[42]. In the presence of an early antigenic stimulus, T cells undergo rapid clonal expansion and high magnitude of IFN-γ response. Precursor cells rapidly differentiate into cytotoxic and exhausted subsets, with low levels of *TCF7* and AP-1 transcriptional factors[24,43]. This rapid response exerts an immunological pressure on the circulating viral populations causing rapid immune escape, in line with previous observations from HCV[44–46] and HIV[47–49]. In contrast, reduced or delayed antigen stimulation results in a decreased probability of differentiation into exhaustion and effector subsets, favouring expansion of memory precursors subsets (T$_{ML}$ and T$_{MEM}$), and retention of AP-1 transcription factors. This model explains the series of unexpected observations made in this study, including the surprising result that high magnitude IFN-γ responses were associated with early onset of exhaustion and with a reduction in AP-1 transcription factors. Finally, ATAC-seq combined with scRNA-seq analysis revealed coordinated changes at the transcriptional and epigenetic levels and provided evidence that precursor of exhausted T cells formed heterogeneous subsets in the early-stage irrespective of disease outcome, and that high functional responses have a distinct epigenetic control, early onset of exhaustion, and decreased accessibility to AP-1 transcription factors.

A key question in T cell biology is how exhaustion is formed and then progresses from the progenitor cells to terminal status[50]. The results of this study revealed that the early onset of exhaustion is a consequence of high magnitude, clonally expanded response, potentially representing a tolerance mechanism to control harmful cytotoxicity and excessive damage to the host[51]. The rapid immune escape resulting from high magnitude IFN-γ responses, leads to increased memory subsets as previously shown[8]. These findings can be explained with a model whereby the maintenance of T cell exhaustion in

the chronic-phase is the result of prolonged responses against conserved epitopes targeted by low-affinity T cell repertoires, as it has been demonstrated in the LCMV model where high-affinity T cells secrete high levels of IFN-γ against early dominant epitopes but are more extensively deleted compared to lower affinity T cells[52]. An alternative scenario of progression of exhaustion is that over the course of the infection, new viral variants arise cyclically[53], thus eliciting new strong responses and leading to similar dynamics that characterise early-stage with rapid onset of exhausted cells. Further studies are needed to better understand the long-term evolution of escape dynamics and how this impact the kinetics of exhaustion. Recent studies on HCV infection revealed that terminally exhausted but not progenitor exhausted cells are lost following clearance with antivirals or loss of antigen due to immune escape, and those that persist carry epigenetic[38] and functional[50,54] scars. Furthermore, transcriptional divergence has been identified as an early marker of cell exhaustion in HCV infection[5,55,56]. Our study revealed that progenitor and early exhausted cells are abundant during early infection, irrespective of disease outcome and that AP-1 transcription factors contribute to differentiation to a terminal exhaustion state. Notably, AP-1 transcription factors are involved in the regulation of T cell activation[57], and downregulation of *FOS* and *JUN* occur in terminally exhausted cells[58]. Our study revealed that progenitors of exhausted cells have a significant reduction in the expression of AP-1 factors since the early-stage of infection, hence suggesting that the fate of exhausted cells is determined early in the infection and before outcome is determined.

Clearance of HCV is known to be associated with an early onset of neutralising antibodies compared to in chronic infection[59–61], however, there is limited knowledge of the activity of neutralising responses targeting the early infecting viruses. HCV antibodies targeting envelope regions of the transmitted founder virus appear at a mean of 71 days post-infection (DPI), and are narrowly directed against the autologous T/F virus, while in individuals progressing to chronic infection, these responses are detected much later[62]. Antibody responses are influenced by the specificity of the infecting virus[63], and viral escape against B cell responses is known to occur during the chronic-phase of infection predominantly associated with rapid emergence of new viral variants[60–62]. These results are consistent with our findings on CD8[+] T cell responses and confirm that early specific responses to transmitted founder variants has an important function in clearance and in determining the probability of onset of immune escape variants.

Our study was limited from the study of earlier time points because HCV-specific CD8[+] T cell responses are absent from the blood in the first three weeks post-infection[64,65]. During this phase, a broader range of T cell repertoires may be recruited to the key site of viral replication in the liver, potentially with different affinities and breadth of responses. We measured IFN-γ secretion to determine functionality, however, other cytokines, such as TNF, may reveal other polyfunctional T cell subsets lacking in IFN-γ production. While polyfunctionality of anti-viral responses is required to better understand the loss of function, the focus of this study was to provide an accurate measure of the IFN-γ magnitude of the CD8[+] T cell response with

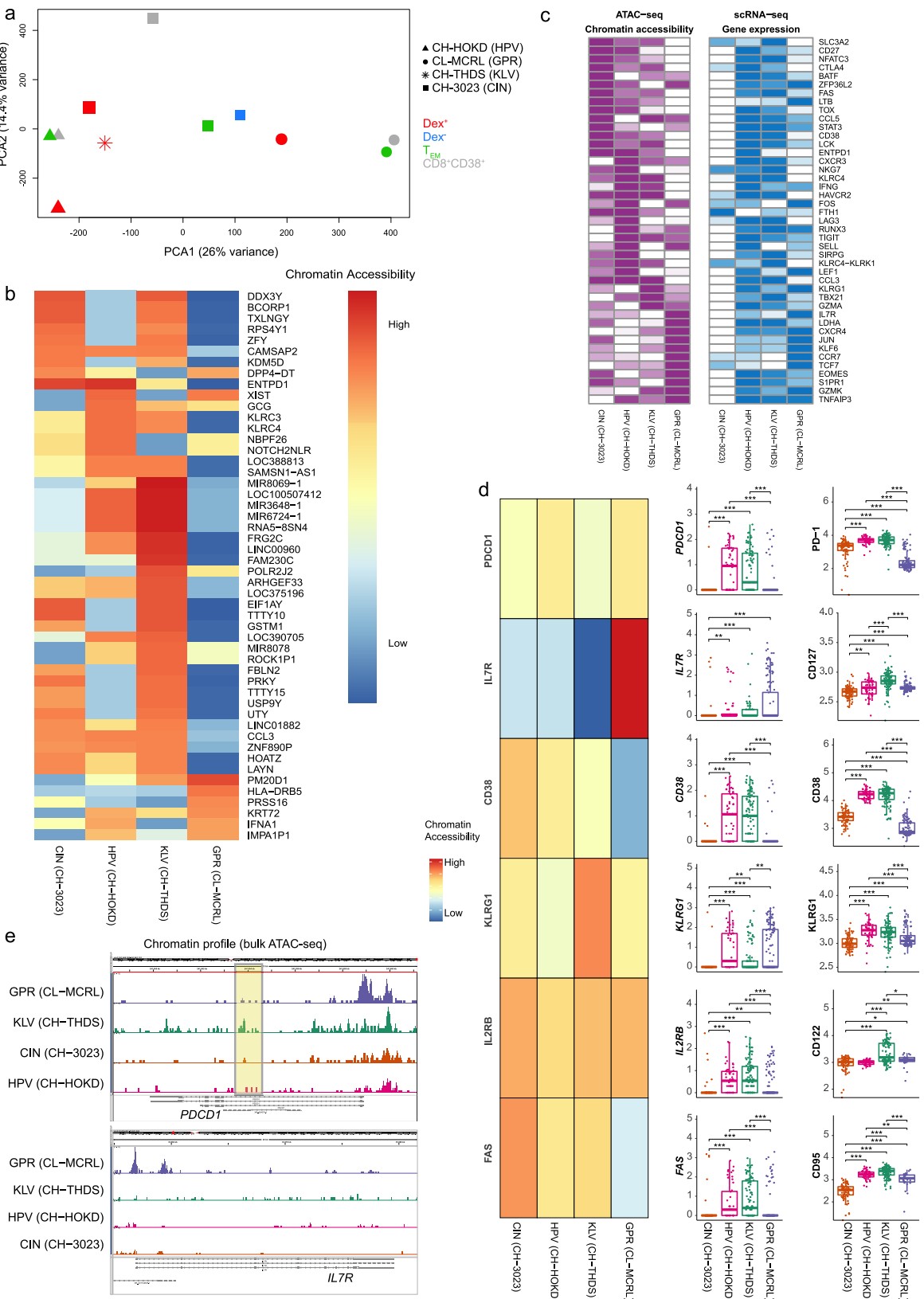

minimal ex vivo perturbations and with an accurate longitudinal analysis of the CD8[+] T cell responses against the earliest autologous infecting variants, before disease outcome is designated. Finally, our findings revealed a differential expression in AP-1 transcription factors in T cell responses associated with clearance and specifically in the early-stage of infection. These findings may be relevant for future

investigations on the early mechanisms that may determine loss of function. For instance, over-expression of c-Jun has been shown to decrease the probability of exhaustion in animal models of CAR T cells[24].

This comprehensive multi-omics and longitudinal analysis of antiviral CD8[+] cytotoxic T cell response in HCV provided further

**Fig. 6 | Heterogenous chromatin accessibility profiles characterise responses targeting conserved and escaping epitopes. a** Principal component analysis (PCA) from chromatin accessible sites of HCV-specific T cell responses (dextramer positive (Dex[+])) and subsets of CD8[+] T cell, from samples within 120 days post-infection. **b** Heatmap of top 50 most variably opened annotated chromatin regions. **c** Traces of chromatin accessibility for the *PDCD1* and *IL7R* loci for each of the four responses. The box drawn in *PDCD1* highlights the opening for the intragenic *PDCD1* cis-element region associated with terminal exhaustion[36]. **d** Chromatin accessibility (left), single-cell gene (middle) and protein (right) expression values for selected markers with available data (N = 369 cells from six biologically independent sample time points across 4 epitope specificities). Pairwise group comparisons were performed with two-sided Wilcoxon Rank Sum Tests (* $p < 0.05$, ** $p < 0.01$, *** $p < 0.001$, **** $p < 0.0001$). Shown are the median and 75% range quantiles. **e** Heatmaps of chromatin accessibility (in purple) and gene expression (in blue) for selected genes. Gene expression is calculated as the average $\log_{10}(TPM + 1)$ of all cells in the early phase (≤120 DPI) of each immune response. Accessibility and expression values are rescaled between [0, 1] per gene, with higher values represented by darker colours.

insights into how exhaustion arises and on the contribution of cytotoxic T cells to viral infection outcome. This knowledge is relevant for designing new vaccines and cellular therapies that elicit a successful and persisting CD8[+] T cell response that limits the rapid onset of immune escape in the context of both virus and cancer. The discovery of the important activity of AP-1 transcription factors should be further validated in other chronic infections and in the context of cancer therapies[66].

## Methods

### Study design
The study aimed to identify the molecular, cellular and phenotypic features of T cell responses in primary HCV infection. Individuals were selected from the Hepatitis C Incidence and Transmission Studies in prisons (HITS-p) and community (HITS-c) cohorts which recruited prospectively followed up high-risk injecting drug users from New South Wales, Australia on the basis of seronegative and HCV RNA-negative tests[15,67,68]. Human research ethics approvals were obtained from Human Research Ethics Committees from the University of New South Wales Human Research Ethics Committee (HC190074). Written informed consent was obtained from the participants. All methods were performed in accordance with the relevant guidelines and regulations. Eligible participants had a lifetime history of injecting drug use and were documented to be anti-HCV and RNA-HCV negative in the 12 months prior to enrolment. Following initial detection of viremia, blood samples were collected frequently over a 24-week period until spontaneous clearance or chronic infection was established. Infection outcome was determined by considering whether each individual continued to test positive for viral RNA at six months following initial detection of viraemia. HCV antibody (Ab) and HCV RNA testing was performed as previously described[69].

Participation in the HITS-p cohort provided benefit to the individual participants by raising awareness of the risk of hepatitis C transmission in the prisons, and the potential benefits of prompt diagnosis. The cohort and its scientific outcomes raised awareness in the health and custodial authorities of the incidence and risk factors for hepatitis C transmissions in the prison setting[15,70].

### Individuals and samples
A total of 17 individuals were selected for this study. Fourteen individuals were analysed for viral sequencing, IFN-γ production, and flow cytometric phenotyping. Six individuals from these 14 were selected for scRNA-seq. An additional three individuals (CH-3221, CH-3132, CH-4059) were analysed by scRNA-seq only. All individuals selected for this study had longitudinally collected samples with estimated days post-infection (DPI) previously described[71] and known infection outcomes. The estimated date of infection for early incident cases was estimated by subtracting the recognised mean pre-seroconversion window period of 51 days from the midpoint between the last HCV RNA[+] HCV Ab[-] time point and the first seropositive time point[17]. For individuals with HCV RNA[+] HCV Ab[+] status at the initial infection time point, the estimated data post-infection was estimated as the midpoint between the last available HCV Ab[-] and the first available HCV Ab[+] timepoint[71]. Blood samples were obtained longitudinally from participants enroled in the HITS-p and HITS-c cohorts. Blood samples were processed for isolation of PBMC using Ficoll-Paque gradients and then cryopreserved. Single-cell multi-omics was performed using cryopreserved PBMC, which were thawed, quality checked for viable and live cells and then stained with dextramers for single-cell sorting.

### Reporting summary
Further information on research design is available in the Nature Portfolio Reporting Summary linked to this article.

## Data availability
Single-cell RNA seq data are available on Accession number GSE196330. All the single-cell RNA-seq data are deposited with GSE196330. All the other data generated and analysed during this study including ATAC-seq, Flow cytometry data as well as viral deep sequencing are all available upon request to the authors. Source data are provided with this paper.

## Code availability
The script files to perform all analysis to reproduce the data/results in the paper as well as recreate all figures are deposited at Zenodo, with the following link: https://doi.org/10.5281/zenodo.7309628.

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

## Acknowledgements

The HITS-p and HITS-c investigators include Andrew Lloyd, Lisa Maher, Kate Dolan, Paul Haber, William Rawlinson, Carla Treloar and Greg Dore. Flow cytometry and sequencing were supported by staff at the UNSW Flow Cytometry and Ramaciotti Centre core facilities. This research was supported from National Health and Medical Research Council of Australia (NHMRC) - Project Nos. APP1121643, 1027551, 1060199, Partnership No. 1016351, and Programme Nos. 510488 and 1053206. The HITS-c cohort was supported by the UNSW Hepatitis C Vaccine Initiative and NHMRC Project Grant No. 630483. F.L., A.R.L. and R.A.B. are supported by NHMRC Research Fellowships (Numbers: 1128416, 1041897 and 1084706). M.R.P., P.L., and C.C. were supported by an Australian Government Research Training Programme (RTP) Scholarship. S.G. was supported by the NHMRC grant APP1148284. K.K. is supported by the NHMRC grant APP1148284. Leadership Investigator Grant (#1173871), T.H.O.N. is supported by NHMRC Emerging Leadership Level 1 Investigator Grant (#1194036).

## Author contributions

F.L. designed and led the study with support from R.A.B. and A.R.L. A.R.L. designed and led the cohort studies. M.R.P. and E.K. performed the ELISpot experiments. C.C., M.R.P., and E.K. performed the flow cytometry experiments. C.C., M.R.P., F.L. and E.K. analysed the ELISpot and flow cytometry data. S.G., P.L., and A.R.L. provided support for the ELISpot, flow and viral epitopes identification analyses. C.C., T.N.A. and A.E. performed the scRNA-seq experiments. S.G., A.R.L., K.K., and T.H.O.N. contributed reagents. J.S., B.H., J.L.P., R.L., M.G., C.C., T.P., S.R., and W.v.d.B. analysed the scRNA-seq data. T.N.A. and C.C. performed the ATAC-seq experiments. J.S. and T.P. analysed the ATAC-seq data. F.L., J.S., and C.C. performed remaining analyses and generated figures. F.L. wrote the manuscript with support from C.C. and J.S. All authors read the manuscript and provided comments.

## Competing interests

The authors declare no competing interests.
