## [Peer Review File · Nature Communications]

Identification of human progenitors of exhausted CD8+ T cells associated with elevated IFN- γ response in early phase of viral infectionREVIEWER COMMENTS

Reviewer #1 (Remarks to the Author):

In this study, the authors were able to investigate a unique cohort of patients from onset of a primary HCV infection. The longitudinal analysis of several HCV epitope-specific CD8 T cells revealed that an elevated IFN γ response during acute phase of HCV infection is associated with a higher rate of immune escape and early onset of exhaustion. Precursor cells thereby rapidly differentiate into exhausted cells with an activated, cytotoxic phenotype. In contrast, a lower IFN γ response favors the expansion of memory precursor subsets and retention of AP-1 transcription factors. The study is of potential interest as the comprehensive multi-omics and longitudinal analysis provide new insights on how exhaustion arises and how cytotoxic CD8+ T cells contribute to disease outcome.

However, there are some points that need clarification before publication:

Major Points:

1. The study has a remarkable cohort of HCV infected patients who were followed longitudinally from onset of the infection. The authors analyzed the phenotype of HCV epitope-specific CD8+ T cells over the time course of infection. The data showed a clear dispersion.

a) Is it possible that the dispersion is caused by combining the different time points during HCV infection? It would be helpful if the authors could please split their data not only by disease outcome but also by early and late phase of infection as well as after resolution or during chronic phase. When are the first differences in the phenotype of HCV epitope-specific CD8+ T cells detectable between subjects who clear the infection versus patients who develop chronic infection? It would be beneficial if the authors would include, in addition to the exhaustion-associated markers, markers for effector differentiation such as Ki67, GrzB, Prf1, CD39 as well as memory associated markers such as TCF1 for their phenotypic analysis.

b) Fig 2A already indicates that there are differences between the individual epitopes using CD127/PD1 co-expression. Could the dispersion of the data possibly be due to the grouping of 20 different epitopes and 3 different genotypes? Since it is already described in the literature for both LCMV and HBV that different epitopes/antigens can cause different T-cell responses, the authors should please comment and discuss this.

2. The authors nicely show in their study that the magnitude of the IFN γ response determines whether viral escape occurs fast or slow and whether the patients are able to clear the virus or develop a chronic infection. Do the authors find other functional features of HCV epitope-specific CD8+ T cells such as TNF production or cytotoxicity (GrzB, Perforin expression) that differ between the two groups?

3. The authors demonstrate in their study that AP-1 transcription factors have an important role in viral clearance. It would be interesting to confirm the results from the scRNAseq data on the protein level? Have HCV epitope-specific CD8+ T cells from patients who clear infection indeed a higher expression of AP-1 transcription factors compared to patients who develop chronic infection?

4. The authors performed a single cell multi-omic approach of their HCV epitope-specific CD8+ T cells obtained from patients who clear the infection versus patient who develop chronic infection. In the dimensionality reduction of the scRNAseq data via UMAP, a very heterogeneous distribution and no real clustering of the cells according to disease outcome or time course of infection was observed. However, a clear difference in gene expression was found between both the CH and CL groups. The authors should comment on why the cells do not cluster according to disease outcome when these cells show such significant differences on gene expression level. The authors should please validate the results on protein level. It would be moreover beneficial if the authors could perform a cluster analysis using Harmony or Race ID to characterize specific clusters of HCV epitope-specific CD8+ T cells.

5. In order to investigate the differences in their differentiation process in patients who clear the infection versus patients who develop chronic infection, a combined trajectory analysis would be beneficial. Is there any node that determines disease outcome? The authors should also perform a diffusion map analysis to investigate the differentiation

process.

Minor Points:

1. The authors should please use a uniform color code for the different T-cell subsets across all figures.
2. The authors should please optimize the quality of the figures, in particular of Fig 6.
3. The authors should please show representative plots to all FACS data.
4. Fig 1A/B: The authors should please insert these two method schemes in the respective figure where the results of these methods were presented. Instead, it would be helpful for the reader to have a scheme of the analysis points post infection in figure 1.
5. Fig 1E: Significance asterisks are missing. The authors should please use a similar illustration of the significance asterisks.
6. Fig 2D: The authors should please not show a summary presentation of their data. It would be beneficial if the authors plot all data points of a single patient and connect the data points over the time course.
7. Fig 2A,B/ Fig4C,D/ Fig7C,D,E: The authors should carefully check the figures for transformation errors in the headings.

Reviewer #2 (Remarks to the Author):

Summary

Cai et al present an in-depth analysis of the function, phenotype and transcriptional profiles of HCV antigen specific CD8+ T cells during primary infection in patients who either experience spontaneous clearance of the virus or chronic infection. This study is a unique opportunity to analyze the CD8+ T cell dynamics in the setting of chronic viral infection in humans, analogous to the heavily utilized murine model of LCMV infection. The authors postulate that a higher initial IFN γ response is associated with immune escape, early exhaustion, and development of chronic viral infection and that heterogeneous subsets of progenitor exhausted T cells undergo distinct differentiation trajectories. The authors support these conclusions with detailed analysis of CD8+ T cell clones by ELISPOT, SMARTseq2 based single-cell RNAseq analysis of HCV specific CD8+ T cells, bioinformatic reconstruction of differentiation and bulk ATACseq. Overall, the work presented here represents an important contribution to the understanding of human CD8+ T cell exhaustion in the setting of chronic viral infection. However, while the authors have done an impressive amount of work and analysis, a careful reading of the manuscript reveals several issues that should be addressed.

Major Comments

1) The authors thoroughly assess the early cellular immune response to HCV in a well-characterized cohort of patients that go on to have chronic infection or viral clearance. This is indeed a useful model in which to study the earliest stages of CD8+ T cell phenotypes in human disease. However, comparing to the Clone 13 versus Armstrong strains of LCMV in mice, interpretation of cellular immune responses to community acquired HCV infection is complicated by the fitness of the transmitted/founder viral quasi-species and the subsequent evolution of the virus over time. To better interpret their CD8+ T cell data, the authors should also evaluate and compare the viral diversity at early timepoints. Do patients in this cohort that go on to have chronic infection have greater viral diversity at early timepoints? What about patients that clear the virus? Analysis of the diversity of the viral sequences would aid interpretation of the CD8+ T cell phenotypes, specifically showing that CD8+ T cell exhaustion is not only a result of increased replication of viral quasispecies. The authors are to be commended for comparing the viral load across their patient cohort, and it is encouraging that there were not substantial differences in viral load at early timepoints.

2) Interpretation of the CD8+ T cell responses in patients with viral persistence versus

those with viral clearance is also complicated by the humoral response to the virus. While HCV antibodies are routinely quantified as positive or negative as part of HCV diagnosis, the anti-viral antibody titers are important for interpretation of the cellular antiviral response. Titers of neutralizing antibodies would be ideal, but beyond the scope of this manuscript. Nevertheless, understanding whether patients with chronic infection versus those with viral clearance have significantly different anti-HCV antibody titers would further aid interpretation of the CD8+ T cell phenotypes. This point and the one above should also be addressed in the discussion to show that there may be other features that contribute to viral clearance beyond CD8+ T cell responses.

3) One of the most exciting aspects of this manuscript is the focus on longitudinal transcriptional analysis of HCV specific CD8+ T cells in the peripheral blood of patients. The authors have extensive experience with transcriptional analysis of T cells and reconstruction of full-length T cell receptor sequences from single-cell RNAseq data. However, careful inspection of the data as presented suggests that technical effects may contribute to some of the differences observed in the current presentation of the data. In Ext Data Fig 2a, it appears that library size and subject substantially contribute to the observed variance. Have the authors evaluated different methods of data integration for their single-cell RNAseq analysis? Since the authors are using Seurat, perhaps it would be most appropriate to evaluate either SCTransform or reciprocal PCA for data integration as implemented in the Seurat workflow. This could potentially reveal alignment across samples that is currently not observed for technical reasons.

4) Another particularly exciting aspect of the manuscript is the pseudotemporal analysis of HCV specific CD8+ T cell states from longitudinally sampled patients. The authors use both PAGA and RNA velocity to infer differentiation trajectories with the PAGA analyses presented in primary figures and the RNA velocity analysis in the supplementary data. While the PAGA graph structure appears informative, some of the pseudotemporal signatures derived from PAGA are difficult to interpret. For example, in Ext Data Fig 4e CL_T2 and CL_T4 appear to be disconnected in the low-dimensional embedding presented. This complicates analysis because it implies two different disjoint cell states rather than a smooth differentiation process as the bioinformatic approach attempts to reconstruct. The same issue is evident in Figure 5f. RNA velocity can also be used to generate a pseudotemporal ordering. Have the authors compared this approach to the results from PAGA? Finally, the authors refer to "families" of differentiation trajectories in several instances, but it is unclear how these families were defined. Is it based on the true notion of a lineage such as those that can be inferred with the Slingshot algorithm, or based on prior biological knowledge?

5) Several recent publications (e.g. Miller et al Nat Immuno 2019; Beltra et al Immunity 2020) have defined hierarchies of progenitor exhausted states in more granular detail than the authors present here. How do the transcriptional profiles of the ML and PINT cells compare to for example the progenitor exhausted states from Beltra et al 2020 Immunity?

Minor comments

1) The authors have appropriately described the processing of their SMARTseq2 data and the generation of TPM normalized counts. However, it is unclear based on the Methods how these counts were used in Seurat. Were the TPM normalized counts log transformed and used for downstream analysis in Seurat, or was Seurat used to normalize for library size? Please describe in the Methods.

2) In the methods, lines 188 – 189 the authors state that the top 7500 genes were used with the VST method in Seurat, while on lines 192-194 they state that the top 3000 variable genes were used for analysis. Please clarify.

3) How were normalized gene counts used for differential expression analysis? The authors cite a method for deconvolution of library size for normalization of gene expression, but they appear to have used a different normalization approach for clustering and UMAP generation in Seurat? Please clarify in the Methods.

- 4) How were genes and gene sets selected for presentation in the figures? Did the authors select biologically meaningful gene or gene sets for presentation? Does preselection of gene sets for presentation bias the analysis? Please describe how subsets of genes or gene sets were selected for analysis in instances where selection was performed.
- 5) On Line 144, the authors refer to Fig 1c. Do they mean Fig 2c?
- 6) In Fig 3c, the authors present dot plots to compare differentially expressed genes between CD8+ T cell subsets. Could a Venn Diagram of differentially expressed genes from each cell subset better show the degree of shared and distinct differentially expressed genes across these populations?
- 7) In Fig 3i, the linear regression line in red has a negative slope, but the correlation coefficient is positive. Fig 3h adequately makes the point better than Fig 3i, which could perhaps be excluded.
- 8) The single-cell RNAseq raw and processed data should be deposited on the Gene Expression Omnibus with an accession number assigned. Currently, the data availability statements have accession numbers listed as "XXX".
- 9) The authors present the ATACseq data in a highly summarized way. It would be useful if the authors presented a more granular analysis of the ATACseq data including differentially accessible regions between samples. This would allow for better interpretation of the data.

Reviewer #3 (Remarks to the Author):

Cai et al. studied longitudinal blood samples with primary infection to analyze the HCV-specific CD8+ T cells. The CD8 T cells was classified into TEX, TML, TPINT, TMEM and TEFf based on the expression level of PD-1 and CD127. The authors claimed they discovered heterogenous subsets of progenitors of exhaustion, based on the level of PD-1 expression and loss of AP-1 transcription factors. They claimed elevated IFN- γ response against the transmitted virus was associated with the rate of immune escape, larger clonal expansion, and early onset of exhaustion. The story is interesting. However, I have a lot technical concerns, which reduce the novelty as well as the major conclusions. In particular, the authors did not show clusters in single cell RNA-seq data that are counterparts of TML and TPINT identified by FACS.

Major concerns:

1) The analysis of scRNA-seq data is not rigorous. Although fig3b showed the expression differences between early phase and late phase at single cell resolution, we could not judge whether these differences were caused by one sample or several samples. We did not know the overall differences between cells from early phase and late phase. Do the cells from early phase and late phase show difference when they were projected on UMAP?

In particular, we do not know how the authors make the threshold on expression of PD-1 and CD127 in scRNA-seq data for classifying the cells into TEX, TML, TPINT, TMEM and TEFf. Could these cell subsets be cluster into groups using scRNA-seq data?

2) The manuscript indicated that the magnitude of IFN- γ responses from CHs were higher than those identified in CLs between 90 and 120 days post-infection (Fig. 1e). However, there should be no significant difference between the two data sets based on the error bar in Fig 1e.

3) The manuscript indicated that CHs had a positive association between the magnitude of response with the proportion of TEX, TML, and activated CD38+ cells (Fig. 2e). In fact, the TEX and TML do not positively associated with magnitude of response in Fig. 2e.

4) The trajectory analysis is also confused me. It is unknown how the trajectories were grouped in two families (4 pseudotime lineages). It will be much better to show the

pseudotime with all cells in fig4 before the authors separated them into two families (4 pseudotime lineages).

5) fig5f, how the cells were classified into two lineages since the overall pseudotime were not provided.

Minor concerns:

1) The garbled codes on figures (fig2a, fig2b, fig7c, fig7d, fig7e) and resolution of fig6 affect reading.

2) fig3c, the colors in figure legend and box plot of TPINT and TMEM are inconsistent. Also a typo TPEX (should be TPINT)

3) The sample numbers described in manuscript are inconsistent with that in figures. e.g..identified in five CLs and seven CHs.

4) "chronic progressor" was abbreviated as CH in most case, while sometimes abbreviated as Ch or even chronic. These abbreviation should be unified.

5) There is no information in Extended data Fig. 2a.

RESPONSE TO THE REVIEWERS' COMMENTS

Reviewer #1 (Remarks to the Author):

In this study, the authors were able to investigate a unique cohort of patients from onset of a primary HCV infection. The longitudinal analysis of several HCV epitope-specific CD8 T cells revealed that an elevated IFN γ response during acute phase of HCV infection is associated with a higher rate of immune escape and early onset of exhaustion. Precursor cells thereby rapidly differentiate into exhausted cells with an activated, cytotoxic phenotype. In contrast, a lower IFN- γ response favors the expansion of memory precursor subsets and retention of AP-1 transcription factors. The study is of potential interest as the comprehensive multi-omics and longitudinal analysis provide new insights on how exhaustion arises and how cytotoxic CD8 $^+$ T cells contribute to disease outcome. However, there are some points that need clarification before publication:

Major Points:

1. The study has a remarkable cohort of HCV infected patients who were followed longitudinally from onset of the infection. The authors analyzed the phenotype of HCV epitope-specific CD8 $^+$ T cells over the time course of infection. The data showed a clear dispersion.

We are pleased to see that this reviewer appreciated the power of studying HCV in this unique cohort of patients. We have now addressed in more detailed the dispersion observed in our data, and not only for the phenotypic data, but also for the molecular (single cell genomics) and functional (IFN- γ ELISpot) data.

a) Is it possible that the dispersion is caused by combining the different time points during HCV infection? It would be helpful if the authors could please split their data not only by disease outcome but also by early and late phase of infection as well as after resolution or during chronic phase. When are the first differences in the phenotype of HCV epitope-specific CD8 $^+$ T cells detectable between subjects who clear the infection versus patients who develop chronic infection?

We agree that the dispersion observed in our functional (ELISpot) and phenotypic (FACS) measures are the result of many factors. These include disease outcome, stage of infection, as well as epitope specificity. To clarify these key results, we have revised Fig. 1 and Fig. 2, and added additional Supplementary figures 1 to 5. We now show the distribution of IFN- γ measures by epitope (Fig. 1g), and by subject and HCV genotype (Fig. S1).

We have modified the text accordingly:

“The IFN- γ SFU values were heterogeneously distributed across epitopes (Fig. 1g) and with no evident separation by disease outcome or HCV genotype (Supplementary Fig. 1c, d).”

Regarding the phenotypic data we have clarified the differences between chronic (CH) and clearers (CL) by disease stage (early vs late) and added a group comparison between early (≤ 120 days post-infection) and late phase (Fig. 2d) as well as using only samples from viremic time points (Fig. S3). These plots now highlight the major early differences in the phenotype between CH and CL, which are: CH have earlier onset of T-bet $^+$ and of T $^{\text{EFF}}$ HCV-specific T cells and less memory cells (T $_{\text{M}}$), with no significant differences in T $_{\text{EX}}$ and markers such as CD160, 2B4, CTLA-4, Tim-3 (Fig. 2d). These results were also confirmed by directly comparing viremic samples across disease outcomes (Fig. S3c). We also noted outliers in the expression of CD160, as well as of CD38 and CD127, which may bias the statistical analysis in the group comparison (Fig. 2d and S3b,c).

We have explained that a significant difference between CH and CL is the decline in effector cells, T-bet $^+$ cells and exhausted (CD127 $^{\text{low}}$ PD-1 $^{\text{high}}$) in CH, while the CD127 $^{\text{low}}$ PD-1 $^{\text{int}}$ (T $_{\text{PINT}}$) remained relatively constant over

time in both disease outcome. To clarify the data further, we have: 1) added a plot in Fig. S3a showing connecting lines between samples of the same epitope response; 2) Retained the original plots with the kinetics of the T cell subsets over time (now in Fig. 2e, Fig. S5). These plots show the estimated rates with which T cell subsets change over the course of the infections (using linear regression).

From Supplementary Figure 3. Analysis of flow cytometry data highlighting epitope and subject specific sample time points.

We have modified the text accordingly:

“Variability between epitopes was evident across all subsets, notably HYP HLA-A*03:01 epitope in CH-3256 expressing high levels of T_{EFF} and CD160⁺ phenotypes (Supplementary Fig. 3a). Analysis of exhaustion and activation markers as well as of transcription factors in the early-stage (<120DPI) revealed increased proportions of T-bet⁺ and of T-bet⁺Eomes⁻ but lower proportions of T_{MEM}, T_{CM} and T_{ML} subsets in CHs (Fig. 2d, Supplementary Fig. 3b). Notably, in this early-stage, CLs and CHs did not differ significantly in the expression of other exhausted associated markers, such as CD160 and 2B4 subsets (Supplementary Fig. 3b). Comparison between CHs and CLs only with viraemic samples confirmed these findings (Supplementary Fig. 3c), thus confirming the increased levels of T-bet in CHs.”

b) It would be beneficial if the authors would include, in addition to the exhaustion-associated markers, markers for effector differentiation such as Ki67, GrzB, Prf1, CD39 as well as memory associated markers such as TCF1 for their phenotypic analysis.

We have clarified that in our phenotypic analysis with FACS we have used two high-dimensional antibody panels for a total of 17 markers with overlapping markers between the two panels (see revised Methods). In addition to the exhaustion markers (CD160, 2B4, CTLA-4, Tim-3, PD-1), we have also utilised activation (CD38), transcription factors (T-bet and Eomes), and effector differentiation (CD45RO, CCR7, CD127, CD27, KLRG1). In the Results section we have clarified this extensive longitudinal phenotypic characterisation of HCV-specific CD8⁺ T cells (Fig. 2), and added additional plots in the Supplementary Figs. 3 and 5.

Regarding the additional markers proposed by this reviewer we were unable to test these markers at the protein level due to a lack of sample availability. We have however added the gene expression values in Fig. 3f and

Supplementary Fig. 6. These additional markers confirm the expression profiles from other genes and also from the FACS profiles, in that T cell subsets have distinct cytotoxicity and differentiation profiles, and that early phase cells from CH have increased levels of differentiation and memory markers.

c) Fig 2A already indicates that there are differences between the individual epitopes using CD127/PD1 co-expression. Could the dispersion of the data possibly be due to the grouping of 20 different epitopes and 3 different genotypes? Since it is already described in the literature for both LCMV and HBV that different epitopes/antigens can cause different T-cell responses, the authors should please comment and discuss this.

We thank the reviewer for this interesting question. Indeed, we have now added additional analysis of the phenotypic data as well as ELISpot data highlighting the dispersion as a function of subject, epitope, and genotype (Fig 1g, Supp. Fig 1c). These figures revealed that the ELISpot values are heterogeneously distributed across epitopes, with some being associated with increased variability/dispersion, and with no evident separation by disease outcome. We have also investigated the distribution of T cell subsets and of IFN- γ response by HCV genotype (Supp. Fig. S1c), which showed no statistical differences.

Given the potential collinearity between epitope and HCV genotype we have also performed a multi-variate analysis to regress out these factors. Firstly, we have performed ANOVA to identify significant factors associated with the heterogeneity (dispersion) T cell subsets across outcomes. This analysis revealed that both HCV genotype and epitope contributed to the observed variability. Epitopes contributed to explain variability in all subsets except T_{PINT}, while HCV genotype explained variability in T_{EFF}, T_M and T_{PINT} subsets (Suppl. Fig. 4). Secondly, we have confirmed these findings by performing a multi-variate regression of T cell subtypes over time, accounting for epitope and HCV genotype as covariates. This analysis confirmed that epitopes were the major factors contributing to the regression of T cell subsets over time. HCV genotype (1b) was significant for exhaustion effector and memory, while 3a only for effector (Supplementary table 7).

We have now discussed these findings in the revised Results section.

“We performed an analysis of variance (ANOVA) to investigate the effect of epitope specificity and HCV genotype on the distribution of T-cell subsets. Both HCV genotype and epitope specificity contributed to the variability observed in each disease outcome (Supplementary Fig. 4). These effects were confirmed by performing a multi-variate regression of T-cell subtypes over time, accounting for epitope and HCV genotype as covariates (Supplementary Table 7). We found that T_{EFF}, T_{PINT} and T_{MEM} revealed genotype specific effects in both outcomes, as well as epitope specific effects (Supplementary Fig. 4). Notably, the genotype 1a was associated with higher proportions of T_{EFF} and lower T_{PINT} in CHs, while in CL, T_{PINT} were higher in genotype 1a compared to genotype 3a. In order to quantify the kinetics of T-cell subsets over the course of the infection we utilised longitudinal data for each epitope specific response and fitted a linear regression model to the experimentally measured proportions of T-cell subsets (Fig. 2e). This analysis showed that a rapid decline of T_{EX}, T-bet⁺, and CD38⁺ subsets in CH, while T_{EFF} increased over the course of infection. In contrast, in CLs no significant decline was observed, with the exception of the PD-1⁺CD38⁺ subset which showed a temporal decline irrespective of disease outcome (Supplementary Fig. 5).”

2. The authors nicely show in their study that the magnitude of the IFN- γ response determines whether viral escape occurs fast or slow and whether the patients are able to clear the virus or develop a chronic infection. Do the authors find other functional features of HCV epitope-specific CD8⁺ T cells such as TNF production or cytotoxicity (GrzB, Perforin expression) that differ between the two groups?

Our focus for this comprehensive analysis is to understand the molecular and phenotypic signatures that determine the magnitude of IFN- γ response. We focussed on extensive investigation of the autologous antigen specific response for each patient and implementing an experimental assay based on a matrix combinatorial approach^{1,2} that maximise the number of epitope specificities that can be tested with limited blood samples, and minimise biases due to ex vivo T cell stimulations, as is the case for assays such as intracellular cytokine

staining. This is an approach used in other viruses and it is well known to be more sensitive than in vitro stimulation³. With this approach we provided an accurate identification of low frequency T cells targeting autologous epitopes without significantly affecting the distribution of T cells or their clonal expansion. Given the uniqueness of our early longitudinal samples, we have opted to investigate in depth the spectrum of epitope specific responses, without exhausting all available PBMC.

We have also added a figure in the Supp. Fig. 6f, showing the expression of several cytotoxicity associated genes in each T cell subsets, revealing the extensive breadth of cytotoxicity profiles, see also the figure below with specific genes associated with cytotoxicity.

Figure R1.1 Gene expression profiles of cytotoxicity-related gene in each T-cell subset.

We have added a Discussion point as follow:

“While polyfunctionality of anti-viral responses is required to better understand the loss of function, the focus of this study was to provide an accurate measure of the IFN- γ magnitude of the CD8⁺ T-cell response with minimal ex vivo perturbations and with an accurate longitudinal analysis of the CD8⁺ T-cell responses against the earliest autologous infecting variants, before disease outcome is designated. ”

3. The authors demonstrate in their study that AP-1 transcription factors have an important role in viral clearance. It would be interesting to confirm the results from the scRNA-seq data on the protein level? Have HCV epitope-specific CD8⁺ T cells from patients who clear infection indeed a higher expression of AP-1 transcription factors compared to patients who develop chronic infection?

We agree with this reviewer that this is an interesting finding. However, as mentioned above, we have already extensively used multiple vials of blood for this study and for these patients it is not possible to perform additional studies due to lack of additional blood samples. We clarified that the scope of this study was to provide an accurate investigation of the molecular, phenotypic and functional features of epitope specific T cell responses in acute phase of infection without ex vivo stimulation-based assays that may impact or bias the

estimate of the true functionality of these cells. These experiments requires significant amount of biological material, and we are confident that our approach is a reasonable and efficient way to maximise the samples we have collated from a unique cohort of injecting drug users followed prospectively. We are grateful that the extensive data we have generated have been recognised and commended by all the 3 reviewers. We have now added this in the Discussion:

“Finally, our findings revealed a differential expression in AP-1 transcription factors in T-cell responses associated with clearance and specifically in the early-stage of infection. These findings may be relevant for future investigations on the early mechanisms that may determine loss of function. For instance, over-expression of c-Jun has been shown to decrease probability of exhaustion in animal models of CAR T cells²⁴.”

Regarding the expression of AP1 transcription factors, we have added a dot plot in Fig. 3h outlining the increased levels of these in CL compared to CH responses, with FOS and revealing additional differenced when early and late phase of infection (<120DPI) are considered. Please refer to the figure below as well. We have also validated with ATAC-seq and confirmed that chromatin accessibility is indeed correlated with gene expression (See Fig. 6c).

Figure R1.2 Comparison of AP-1 transcription factor expression profiles between disease outcome and stage of infection.

4. The authors performed a single cell multi-omic approach of their HCV epitope-specific CD8+ T cells obtained from patients who clear the infection versus patient who develop chronic infection. In the dimensionality reduction of the scRNAseq data via UMAP, a very heterogeneous distribution and no real clustering of the cells according to disease outcome or time course of infection was observed. However, a clear difference in gene expression was found between both the CH and CL groups.

The authors should comment on why the cells do not cluster according to disease outcome when these cells show such significant differences on gene expression level. The authors should please validate the results on protein level. It would be moreover beneficial if the authors could perform a cluster analysis using Harmony or Race ID to characterize specific clusters of HCV epitope-specific CD8+ T cells.

To clarify these comments as well as the ones raised below by the other reviewers, we have re-analysed the scRNAseq and index sorting data (protein expression). We have added a clustering analysis (new Fig. 3). This analysis also accounted for batch effect and have utilised the method implemented in Seurat based on SCTransform and data integration algorithm (see Methods and answer to Reviewer 2).

This analysis revealed 10 clusters, and we found that disease outcome (CH or CL), magnitude of IFN- γ response, disease outcome, and stage of infection (early to late) all contributed to explain the observed molecular

heterogeneity in the UMAP clusters, thus confirming the functional (Fig.1) and phenotypic (Fig. 2) findings. Cells from the early-stage of infection that were associated with high IFN- γ response in CH were located on one extreme of the UMAP graph, while in contrast cells from late-stage and low IFN- γ magnitude responses in CL were mostly found on the opposite side of the UMAP. This analysis is shown on Fig. 3 and Supp. Fig. 6. Finally, regarding the validation with proteins index sorted data, we have added in Fig. 3d the expression of proteins in individual cells using index sorting protein data (see also figure below).

Figure R1.3 Validation of scRNA-seq data with single cell protein expression data obtained from index sorting.

5. In order to investigate the differences in their differentiation process in patients who clear the infection versus patients who develop chronic infection, a combined trajectory analysis would be beneficial. Is there any node that determines disease outcome? The authors should also perform a diffusion map analysis to investigate the differentiation process.

We have included a trajectory analysis from the full scRNA-seq data sets (CH+CL) (See Supp. Fig. 6f). For this analysis we have utilised Slingshot on the UMAP latent space and chosen as root cells from cluster 5, which is formed by cells from the earliest time point. We have also validated these findings by choosing as root cluster 3 or 4 with similar results. We have added this result in the revised manuscript:

“To investigate in more details the molecular differences observed in the UMAP and clustering analyses between outcomes and between the early- and late-stages of infection, we reasoned that distinct differentiation trajectories underline the dynamics of HCV-specific T-cell responses in each disease outcome. Firstly, we performed Slingshot trajectory analysis using the UMAP clusters (as in Fig. 3) to generate pseudotime ordering of cells (Supplementary Fig. 6f). We selected cluster 5 as a root, as it was comprised of cells from early sample time points, and identified four trajectories. Two trajectories terminated in clusters formed by late-stage cells (7 and 8), and two remained within early-stage clusters, terminating in clusters 3 and 5, respectively. The two

trajectories ending in clusters 7 and 8 were consistent with a differentiation process from an early activated state into an effector (cluster 7) or memory state (cluster 8), while the other two trajectories were consistent with differentiation into activated and proliferating cells, in line with the phenotypic distribution of early-stage clusters. This analysis revealed that molecular heterogeneity of the HCV T-cell responses is consistent with more than one differentiation process, and that functional state and stage of infection are important factors associated with each differentiation trajectory.”

In the original submission we have already performed a diffusion map analysis using PAGA and pseudotime analysis for CH only, CL only and for the monoclonal T cell response identified in subject MCRL. We have now repeated this analysis for all the data set, as well as for the other subsets using SCTransform normalised data. These new analyses largely confirmed our previous findings, where we discovered heterogenous differentiation between disease outcome and also between responses with differing magnitude of IFN-g response. We have clarified in the results section the reasons supporting a more in-depth trajectory analysis using PAGA and split by disease outcome (Fig. 4, for CH, Supp Fig. 10 for CL). Notably, we have also highlighted the opportunity to perform an intra-clonal fate mapping of a single lineage identified from the monoclonal response (identical TCR clone) in the high IFN- γ GPR-specific T cells in subject (CL-MCRL) (Fig. 5), which was based on 244 cells from 4 samples, two of which were early viraemic samples (day 88 and day 116) and two post-clearance.

Minor Points:

b. The authors should please use a uniform color code for the different T-cell subsets across all figures.

We have checked and confirm that we have already used a uniform color code for T cell subsets. Now we have also used these colours for the labels of the axis of dotplots.

2. The authors should please optimize the quality of the figures, in particular of Fig 6.

We have corrected the colours in legends and improved Fig. 6 containing ATAC-seq results.

3. The authors should please show representative plots to all FACS data.

We have added additional FACS data on dextramer staining and gating of additional markers: T-bet Eomes, CD45RA CCR7, CTLA_4, in the Supplementary Fig. 2.

4. Fig 1A/B: The authors should please insert these two method schemes in the respective figure where the results of these methods were presented. Instead, it would be helpful for the reader to have a scheme of the analysis points post infection in figure 1.

We have now added a scheme of the analysis point in Fig. 1, and retained an overall scheme for the experimental design and overall technology. We have added a scheme of the method used in Fig. 3 on the single cell multi-omics study design, and also in Fig. 4, outlining the strategy to identify trajectories from the PAGA analysis using longitudinal scRNA-seq and index-sorting protein data.

5. Fig 1E: Significance asterisks are missing. The authors should please use a similar illustration of the significance asterisks.

We have now replaced asterisks with the actual p-values.

6. Fig 2D: The authors should please not show a summary presentation of their data. It would be beneficial if the authors plot all data points of a single patient and connect the data points over the time course.

Given the dispersion of the data, connecting the sample time per points per subject for each of the panels in Figs. 1 and 2 would be very confusing. Instead, we have provided a few examples only, where it can be observed the evolution of these subsets over time. These are now in the Supplementary Fig. S3.

7. Fig 2A,B/ Fig4C,D/ Fig7C,D,E: The authors should carefully check the figures for transformation errors in the headings.

We have revised Figures 2 and 4 and removed the original Figure 7.

Reviewer #2 (Remarks to the Author):

Summary

Cai et al present an in-depth analysis of the function, phenotype and transcriptional profiles of HCV antigen specific CD8+ T cells during primary infection in patients who either experience spontaneous clearance of the virus or chronic infection. This study is a unique opportunity to analyze the CD8+ T cell dynamics in the setting of chronic viral infection in humans, analogous to the heavily utilized murine model of LCMV infection. The authors postulate that a higher initial IFN γ response is associated with immune escape, early exhaustion, and development of chronic viral infection and that heterogeneous subsets of progenitor exhausted T cells undergo distinct differentiation trajectories. The authors support these conclusions with detailed analysis of CD8+ T cell clones by ELISpot, SMARTseq2 based single-cell RNAseq analysis of HCV specific CD8+ T cells, bioinformatic reconstruction of differentiation and bulk ATACseq. Overall, the work presented here represents an important contribution to the understanding of human CD8+ T cell exhaustion in the setting of chronic viral infection. However, while the authors have done an impressive amount of work and analysis, a careful reading of the manuscript reveals several issues that should be addressed.

Major Comments

1. The authors thoroughly assess the early cellular immune response to HCV in a well-characterized cohort of patients that go on to have chronic infection or viral clearance. This is indeed a useful model in which to study the earliest stages of CD8+ T cell phenotypes in human disease. However, comparing to the Clone 13 versus Armstrong strains of LCMV in mice, interpretation of cellular immune responses to community acquired HCV infection is complicated by the fitness of the transmitted/founder viral quasi-species and the subsequent evolution of the virus over time. To better interpret their CD8+ T cell data, the authors should also evaluate and compare the viral diversity at early timepoints. Do patients in this cohort that go on to have chronic infection have greater viral diversity at early timepoints? What about patients that clear the virus? Analysis of the diversity of the viral sequences would aid interpretation of the CD8+ T cell phenotypes, specifically showing that CD8+ T cell exhaustion is not only a result of increased replication of viral quasispecies. The authors are to be commended for comparing the viral load across their patient cohort, and it is encouraging that there were not substantial differences in viral load at early timepoints.

We thank this reviewer for the insightful comment on the role of the transmitted founder virus (TF). Our group has pioneered the investigation of the TF in HCV^{1,5}. In these previous papers, we have shown that within the first 3 months post infection, the viral quasispecies first increase and then rapidly decrease at about 100 days, thus determining a genetic bottleneck. We then demonstrated that during this early phase, CD8 T cell responses have a peak in IFN- γ production, hence suggesting a role of CD8 T cell in the selective force driving genetic bottleneck. We also showed that following this genetic bottleneck, at about 100 days post-infection, viral quasispecies rapidly evolve and new strains replace the founder virus carrying mutations that are dominant (fixation events). We showed that many of these are escape variants and are present within epitope targets. In these papers we also showed that in clearers the viral population do not acquire fixation events, while these often occur in chronic patients.

To corroborate further these previous findings, we have now included a plot comparing viral diversity, measured as Shannon entropy value across the full genome of the virus, between chronic and clearers, and showing data for early phase (less than 100 days, and less than 120 days) (Fig. 1, Fig. S1). In both of these we did not observe any statistical difference, hence suggesting that viral diversity does not explain onset of rapid IFN- γ response and early exhaustion. This is indeed a confirmation of our result, in that high IFN- γ response occur in chronic more than in clearers, and that this is likely to impinge upon the viral quasi-species, leading to fixation events via specific mutations within epitope regions. We hope that this revised figure 1 will provide a clearer and more convincing message.

2) Interpretation of the CD8+ T cell responses in patients with viral persistence versus those with viral clearance is also complicated by the humoral response to the virus. While HCV antibodies are routinely quantified as positive or negative as part of HCV diagnosis, the anti-viral antibody titers are important for interpretation of the cellular antiviral response. Titers of neutralizing antibodies would be ideal, but beyond the scope of this manuscript. Nevertheless, understanding whether patients with chronic infection versus those with viral clearance have significantly different anti-HCV antibody titers would further aid interpretation of the CD8+ T cell phenotypes. This point and the one above should also be addressed in the discussion to show that there may be other features that contribute to viral clearance beyond CD8+ T cell responses.

We agree with this reviewer and stress the point that limited studies exist on the role of HCV antibodies in acute phase of HCV infections as the majority of studies focussed on these responses after clearance or in chronic phase. We have recently reported of an investigation of Envelope-specific antibodies targeting transmitted founder viruses ⁶.

We added this paragraph to the Discussion

“Clearance of HCV is known to be associated with early onset of neutralising antibodies compared to CH⁵⁹ ⁶⁰. ⁶¹. However, there is limited knowledge on the role of neutralising responses targeting the early infecting viruses. HCV antibodies targeting envelope regions of the transmitted founder virus appear at a mean of 71 days post-infection (DPI), and narrowly directed against the autologous T/F virus, while in subjects progressing to chronic infection these responses are detected much later ⁶². Antibody responses are influenced by the specificity of the infecting virus ⁶³ and viral escape against B cell responses are known to occur during chronic-phase of infection predominantly associated with rapid emergence of new viral variants ^{60, 61, 62}. These results are consistent with our findings on CD8+ T-cell responses and confirm that early specific responses to transmitted founder variants play an important role in clearance and in determining the probability of onset of immune escape variants.”

3) One of the most exciting aspects of this manuscript is the focus on longitudinal transcriptional analysis of HCV specific CD8+ T cells in the peripheral blood of patients. The authors have extensive experience with transcriptional analysis of T cells and reconstruction of full-length T cell receptor sequences from single-cell RNAseq data. However, careful inspection of the data as presented suggests that technical effects may contribute to some of the differences observed in the current presentation of the data. In Ext Data Fig 2a, it appears that library size and subject substantially contribute to the observed variance. Have the authors evaluated different methods of data integration for their single-cell RNAseq analysis? Since the authors are using Seurat, perhaps it would be most appropriate to evaluate either SCTransform or reciprocal PCA for data integration as implemented in the Seurat workflow. This could potentially reveal alignment across samples that is currently not observed for technical reasons.

We thank the reviewer for acknowledging the importance of our longitudinal analysis and for the suggestions on the bioinformatics methods. We have performed additional analysis to address the potential technical noise on the interpretation of the scRNA-seq data. We have complemented our original gene expression normalisation

and scaling based on SCRAN, with a second analysis based on SCTransform as well as SCTransform in combination with data integration for batch correction implemented in Seurat. To improve clarity in our narrative, we have now presented all our scRNA-seq analyses utilising the SCTransform approach (see revised Methods). In the revised Fig. 3 we have shown the results of batch corrected scRNA-seq data, which largely confirm our main original findings and clarified further the relationship between magnitude of IFN- γ , disease stage and outcome. By normalising and batch correcting our data, we now show how that epitope specificity and subject are not significantly affecting clustering, and reveal a more uniform distribution of cells from individual T cell responses in each cluster.

Figure R2.1 Results from normalisation and batch correction of scRNA-seq data utilising SCTransform.

4) Another particularly exciting aspect of the manuscript is the pseudotemporal analysis of HCV specific CD8+ T cell states from longitudinally sampled patients. The authors use both PAGA and RNA velocity to infer differentiation trajectories with the PAGA analyses presented in primary figures and the RNA velocity analysis in the supplementary data.

a. While the PAGA graph structure appears informative, some of the pseudotemporal signatures derived from PAGA are difficult to interpret. For example, in Ext Data Fig 4e CL_T2 and CL_T4 appear to be disconnected in the low-dimensional embedding presented. This complicates analysis because it implies two different disjoint cell states rather than a smooth differentiation process as the bioinformatic approach attempts to reconstruct. The same issue is evident in Figure 5f.

We are pleased that this reviewer found our longitudinal analysis with single cell multi-omics a novel and interesting step. For consistency with the revised analysis using SCTransform (instead of SCRAN), we have recomputed the PAGA and trajectories analysis and reconstructed pseudotime trajectories with a more stringent set of conditions. Trajectories are now based on full connectivity between clusters (identified with PAGA), as well as with trajectory roots chosen within clusters comprising cells from early sample time point. Each trajectory terminates with a fully connected cluster that comprise cells predominantly from late cells. This set of criteria has been now further detailed in the revised manuscript and have been chosen to take advantage of longitudinal information. All trajectories are now describing a smoothed differentiation process. The new trajectories confirm the major findings of this study which is to identify differentiation trajectories that explain magnitude of IFN- γ responses, onset of immune escape as well as the temporal stage of the infection (See revised Fig. 4, 5 and associated Supp. Fig. 9 and 10).

b. RNA velocity can also be used to generate a pseudotemporal ordering. Have the authors compared this approach to the results from PAGA?

We have reperformed this analysis with the new SCTransform matrix and confirmed the findings in the original submission, whereby RNA velocity results are in line with the trajectory inference based on diffusion map based on PAGA, but not in the directionality we have identified based on the longitudinal data sampling. In

some instances, RNA velocity vectors pointed towards early clusters. Therefore, we have placed a greater priority on our known metadata information, namely days-post infection. For example, from the analysis of the monoclonal response GPR specific (Fig. 5) or for CL, we obtained the following RNA velocity estimates:

Figure R2.2 Analysis of RNA velocity using scVelo.

Investigation of this result revealed that this contrasting directionality with same pattern of connectivity was largely due to the fast RNA velocity values in cells from early phase of infection, which is largely determined by the ratio of spliced over unspliced RNA variants of the same gene. This means that in our data, the RNA velocity inference of direction pointed towards early cells rather than late ones. We have also searched literature and found similar scenarios, including recent works cautioning on the interpretation of RNA velocity data, including from the authors of the scVelo package, and by mathematicians in (Zheng et al. <https://www.biorxiv.org/content/10.1101/2022.06.19.494717v1>) and <https://www.biorxiv.org/content/10.1101/2022.02.12.480214v1.full.pdf>.

We believe that RNA velocity in the context of this manuscript would not be the ideal approach as our data may have timepoints too far apart for which splicing information is no longer informative. Indeed, the original RNA velocity manuscript also states (Lo Manno et al. Nature) “predicts the future state of individual cells on a timescale of hours”, whereas our data have timepoints spread over at least several months.

c. Finally, the authors refer to “families” of differentiation trajectories in several instances, but it is unclear how these families were defined. Is it based on the true notion of a lineage such as those that can be inferred with the Slingshot algorithm, or based on prior biological knowledge?

We have removed the definition of family of trajectory and simplified the interpretation of the PAGA analysis. For instance, in the revised Fig. 4 we have considered 3 trajectories (CH-T1, CH-T2, CH-T3) that can be obtained from the PAGA analysis of CH cells and showed that these trajectories define distinct dynamics. CH-T2 identified the differentiation from early cells rapidly acquiring an effector phenotype that were associated with high magnitude IFN- γ production (see Loess curves, Fig 4). In the revised Result section, we have revised the reasons supporting a more in-depth trajectory analysis using PAGA and split by disease outcome. Briefly, we have now first performed a trajectory analysis using SLINGshot based on the full data sets and the UMAP structure (Supp. Fig. 6) (See also response to the reviewer 1 question 5). We have then moved to a more detailed analysis using PAGA where the inference of trajectory was based on the estimated connectivity between clusters (from PAGA) and the prior biological knowledge available, which is the longitudinal information of the samples.

This reviewer correctly points out on the important concept of true lineage versus inferred trajectory from gene expression dimensionality reduction, such as Slingshot. We would like to point out that both SLINGSHOT and

PAGA are computational algorithms to infer differentiation between cells based on the transcriptomics data, however these do not account for the actual time course, nor of phenotypic differences that may be known, and notably do not account for intra-clonal differentiation. The latter is probably a key information for more accurate inference of T cell differentiation, as it accounts for clones that may have different affinity for the antigen or a different starting point during the infection. In this regard, we have now highlighted further our serendipitous discovery of a monoclonal (identical full-length TCR $\alpha\beta$) response in subject CL-MCRL targeting GPR- HLA-B*-0702 epitope. In this intra-clonal trajectory analysis we identified two distinct lineages, one of which was characterised by high cytotoxicity gene signature, largely explaining the high magnitude IFN- γ associated with this response, and a second revealing memory cells with sustained level of progenitor cells (Fig. 5).

5) Several recent publications (e.g. Miller et al Nat Immuno 2019; Beltra et al Immunity 2020) have defined hierarchies of progenitor exhausted states in more granular detail than the authors present here. How do the transcriptional profiles of the ML and PINT cells compare to for example the progenitor exhausted states from Beltra et al 2020 Immunity?

We have compared our scRNAseq data with the mouse data that have been used in Beltra et al. and in Miller et al. Firstly, we used Miller data as a reference and calculated similarities scores with our data set using SingleR. This analysis revealed that most of our cells clustered within the “Progenitor of exhaustion” and with the “Proliferating” subsets, see below.

Figure R2.3 Reference based analysis of Miller et al data set.

We have also tested the four subsets reported in Beltra et al. we have calculated the expression levels of the genes representing the 10 clusters reported in Table S3 and Fig. 4D of Beltra et al. (see below). There was no clear match between our 5 T cell subsets and the 10 clusters signatures, see below.

Figure R2.4 Gene module scores based on the 10 cluster signatures identified in Beltra et al.

We therefore concluded that our gene signatures from human subsets in early phase of HCV infections are not directly comparable with the mouse data. We noticed that AP-1 transcription factors were not part of the mouse

gene signatures. We compared selected transcription factors across our 5 T cell subsets, which confirmed that T_{ML} and T_{PINT} and T_{ML} are consistent with an intermediate stage of differentiation, expressing high level of *TBX21* (as in Beltra et al Module 5). However, we also found that AP-1 transcription factors featured a progressive loss from T_{ML} to T_{PINT} and finally T_{EX} . See below

Figure R2.5 Expression profiles of transcription factors in each T cell subsets identified in this study.

Finally, we have also compared our data with human exhaustion signatures derived from TILs in liver cancer from Zheng et al. Cell 2017¹¹. This is a commonly used T cell exhaustion signature which revealed that T_{EX} were closely related to the original subset of exhausted cells, while T_{PINT} and T_{ML} overlapped with the gene signatures of T cells identified in the original study as effector as well as $CD8^+LEF1^+$ cells identified as resting cells (naïve or memory).

Figure R2.6 Reference-based comparison with single cell data from human TILs (Zheng et. al.)

This analysis led us to the conclusion that in early phase of HCV infection heterogeneous progenitor subsets can be identified that only partially overlap with known mouse signatures and with terminally exhausted cells commonly found in TILs. In the revised manuscript, we have included the analysis with the Miller et al and Zheng et al. data sets, and discussed the differences we observed between our data and the mouse progenitor populations and the human data.

We have added this section to the Results:

“We next compared the gene signatures of progenitors of exhausted cells T_{PINT} and T_{ML} with those previously identified in the mouse model of LCMV infection^{27, 28}, as well as with T-cell exhaustion profiles from human tumour infiltrating lymphocytes²⁹. We annotated our scRNA-seq data with the reference data set on progenitor cells from Miller et al.²⁸ (Supplementary Fig. 8), and found that T_{PINT} and T_{ML} were mostly correlated with the subset of progenitor of exhausted cells, while T_{EX} revealed a correlation with both progenitor and terminally exhausted subsets. Notably T_{PINT} and T_{ML} retained higher levels of expression of AP-1 transcription factor when

compared to T_{EX} (Supplementary Fig. 7b). Comparison with human exhaustion signatures derived from TILs in liver cancer revealed that T_{EX} were closely related to the original subset of exhausted cells, while T_{PINT} and T_{ML} overlapped with the gene signatures of T cells identified in the original study as effector as well as $CD8^{+}LEF1^{+}$ cells identified as resting cells (naïve or memory).²⁹. This analysis revealed that T_{PINT} and T_{ML} are both consistent with known profiles of progenitor of exhausted cells, however these differ from the subsets identified in the mouse model of LCMV, are not terminally differentiated exhausted cells.”

Minor comments

1) The authors have appropriately described the processing of their SMARTseq2 data and the generation of TPM normalized counts. However, it is unclear based on the Methods how these counts were used in Seurat. Were the TPM normalized counts log transformed and used for downstream analysis in Seurat, or was Seurat used to normalize for library size? Please described in the Methods.

We have rewritten this method section and clarified that we have used TPM counts in Seurat, and how these have been used for downstream analysis in Seurat. As outlined in the comment 3 above, we have now normalised TPM values using SCTransform and used these for downstream analyses.

2) In the methods, lines 188 – 189 the authors state that the top 7500 genes were used with the VST method in Seurat, while on lines 192-194 they state that the top 3000 variable genes were used for analysis. Please clarify.

We have now clarified this step and removed the limit of 7500 genes that was set with the original approach. In the revised analysis we have used all the genes and used in the FindVariableFeature command in Seurat. The top 3000 variable genes were used to run PCA, prior to UMAP calculation.

3) How were normalized gene counts used for differential expression analysis? The authors cite a method for deconvolution of library size for normalization of gene expression, but they appear to have used a different normalization approach for clustering and UMAP generation in Seurat? Please clarify in the Methods.

We clarified this section in the method and added more details on the differential gene expression analysis which was performed using MAST. Please note that in the revised manuscript we have removed the SCRAN normalisation and used the SCTransformed method. All our analyses are now consistent and have used the same normalisation.

4) How were genes and gene sets selected for presentation in the figures? Did the authors select biologically meaningful gene or gene sets for presentation? Does preselection of gene sets for presentation bias the analysis? Please describe how subsets of genes or gene sets were selected for analysis in instances where selection was performed.

We have now clarified that the selection of genes has been performed by accounting for significance of the result of DEG (p-value and fold change, as explained in the methods) as well as by investigating the GSEA enrichment scores associated with these genes. Given the large list of genes fulfilling these criteria, we have selected representative genes based on the associated function. This decision was made purely for visualization and hence biological relevant or known genes were selected. To not bias the results, we have reported all the identified genes and gene signatures in the Supplementary table 8.

5) On Line 144, the authors refer to Fig 1c. Do they mean Fig 2c?

Yes , we have now corrected this typo.

6) In Fig 3c, the authors present dot plots to compare differentially expressed genes between CD8+ T cell subsets. Could a Venn Diagram of differentially expressed genes from each cell subset better show the degree of shared and distinct differentially expressed genes across these populations?

We have now added an upset plot, which is similar to the Venn Diagram, and reported the number of differentially expressed genes (up or down regulated with adjusted p-value < 0.05) between T cell subsets (Fig 3). We have retained the dot plot to identify few key genes to distinguish the subsets.

7) In Fig 3i, the linear regression line in red has a negative slope, but the correlation coefficient is positive. Fig 3h adequately makes the point better than Fig 3i, which could perhaps be excluded.

Fig 3i report the result of a linear regression and we have shown the R^2 value. This is explained in the text and in the legend of Fig. 1. We believe this regression is a distinct finding from the group comparison shown in 3h, as this outline the continuous relationship between TCR diversity and IFN- γ magnitude of response, which is a point we would like to highlight also with our trajectory analysis in the subsequent figure.

8) The single-cell RNAseq raw and processed data should be deposited on the Gene Expression Omnibus with an accession number assigned. Currently, the data availability statements have accession numbers listed as “XXX”.

Single cell RNA seq data are available on Accession number GSE196330.

9) The authors present the ATACseq data in a highly summarized way. It would be useful if the authors presented a more granular analysis of the ATACseq data including differentially accessible regions between samples. This would allow for better interpretation of the data.

We have revised Fig. 6 and added more details on the chromatin profiles identified in the HCV-specific T cell responses analysed. We show uploaded a supplementary table containing all the normalised peaks that have been used to identify the most variable chromatin accessibility peaks (Supplementary Table 12), and compared these against the gene expression scRNA-seq data. We have also reported differentially open peaks when we compared the tetramer positive responses against effector CD8+ T cell. Finally, we have clarified the data interpretation in the Results section.

Reviewer #3 (Remarks to the Author):

Cai et al. studied longitudinal blood samples with primary infection to analyze the HCV-specific CD8+ T cells. The CD8 T cells was classified into TEX, TML, TPINT, TMEM and TEFB based on the expression level of PD-1 and CD127. The authors claimed they discovered heterogenous subsets of progenitors of exhaustion, based on the level of PD-1 expression and loss of AP-1 transcription factors. They claimed elevated IFN- γ response against the transmitted virus was associated with the rate of immune escape, larger clonal expansion, and early onset of exhaustion. The story is interesting. However, I have a lot technical concerns, which reduce the novelty as well as the major conclusions. In particular, the authors did not show clusters in single cell RNA-seq data that are counterparts of TML and TPINT identified by FACS.

We have now provided additional clustering analysis of scRNA-seq data (see also response to reviewer 1 as well), and revised Fig. 3. Specifically, we clarified that single cell data revealed additional heterogeneity as T_{ML} and T_{PINT} were found in several clusters, and in both early and late phase of infection, as well as in both outcomes. For instance, clusters 3, 4, and 5 correspond to early sample time points and each carrying 20-25% of cells that are classified as T_{PINT} . These results revealed that the molecular heterogeneity between cells is consistent with the classification of T cells by CD127 and PD1 values, however cluster gene signatures revealed

additional molecular heterogeneity, thus highlighting the power of single cell multi-omics. The 5 phenotypic subsets were identified by index sorting protein expression values from the single cell multi-omics data, and these were consistent with independent FACS data in Fig. 2. Single cell clustering revealed novel molecular signature and unexpected heterogeneity between T cell subsets over the course of infection. We have addressed below the technical concerns of this reviewer.

Major concerns:

1) The analysis of scRNA-seq data is not rigorous. Although fig3b showed the expression differences between early phase and late phase at single cell resolution, we could not judge whether these differences were caused by one sample or several samples. We did not know the overall differences between cells from early phase and late phase. Do the cells from early phase and late phase show difference when they were projected on UMAP?

We have now clarified the results of our scRNA-seq analysis in Figs 3 and associated supplementary figures 6 and 7. The revised batch clustering and dimensionality reduction analyses based on SCTransform clarify the molecular differences between cells sampled in early and late phase of infection, and across multiple samples and epitope specificity (see UMAP plots in Fig. 3 and Supplementary Fig. 6, colour coded by phenotypic and sample characteristics). In Fig.3h we have added a dot plot showing the result of differential gene expression analysis considering early and late phase cells and across both disease outcomes, CH ad CL.

“Comparisons across the early- and late-stages of infection showed that cells from CHs had early expression of genes associated with exhaustion (*PDCD1*, *ENTPD1*), and cytotoxicity, including NK-like genes *NKG7*, *GZMB*, *KLRC2* and *KLRC3* (encoding for NKG2C and NKG2E, respectively) (Fig. 3h). These cells also expressed higher levels of *TBX21*, *TOX*, and activation markers *CD38* and *HLA-DR* (Supplementary Table 9). In contrast, CLs in the early-stage had a higher expression of transcription factors *FOS* and *JUN*, and memory-like associated genes (*IL7R*, *TCF7*, *CCR7*), which also persisted in the late-stage of infection, where cells had higher expression of transcription factor *BACH2* and of *SIPR1* (encoding sphingosine-1-phosphate receptor-1), which is known to promote T-cell retention in non-lymphoid tissues³⁰. GSEA analysis revealed an enrichment of oxidative phosphorylation, proliferation, cytotoxicity, NK-like and exhaustion gene signatures in early-stage CHs, while cells in CLs showed a sustained enrichment of progenitor of exhaustion, and memory profiles (Supplementary Fig. 7c, Supplementary Tables 10, 11).”.

In particular, we do not know how the authors make the threshold on expression of PD-1 and CD127 in scRNA-seq data for classifying the cells into T_{EX}, T_{ML}, T_{PINT}, T_{MEM} and T_{EFF}. Could these cell subsets be cluster into groups using scRNA-seq data?

We have clarified that classification of cells by phenotype was based on single cell MFI protein index sorting values associated with the scRNA-seq data. These were consistent with the results from the previous separate experiment where we phenotyped cells using FACS (Fig. 2, independent experiments). The threshold for identifying these subsets in both index sorting (single cell multi-omics Fig. 3) and FACS experiments (Fig. 2) were based on same criteria, namely manual gating and FMO values for PD-1 and CD127. We have clarified this in the Result section:

“As T-cell subsets were distributed across all clusters, we reasoned that additional heterogeneity could be identified between the five T-cell subsets. Firstly, we confirmed consistency between gene and protein expression, by comparing The mean fluorescence intensity (MFI) index sorting values with the corresponding gene expression, for all the available pairs, including CD127, PD-1, CD38 and CD95 (Fig. 3d, Supplementary Fig. 6c).”

And in the methods

“The mean fluorescence intensity (MFI) values obtained from the index sorting were used to identify protein expression values of individual cells and to validate scRNA-seq gene expression profiles”.

Regarding the molecular heterogeneity of the T cell phenotypes we have clarified that these subsets were found across all clusters (Fig. 3c). See also response to reviewer 2 Fig. R2.1. To clarify further the molecular signatures, we calculated differentially expressed genes and reported the results in Fig. 3f.

2) The manuscript indicated that the magnitude of IFN- γ responses from CHs were higher than those identified in CLs between 90 and 120 days post-infection (Fig. 1e). However, there should be no significant difference between the two data sets based on the error bar in Fig 1e.

We have tested this comparison using a Wilcoxon sum rank test in R and in Prism and the results are both significant.

3) The manuscript indicated that CHs had a positive association between the magnitude of response with the proportion of TEX, TML, and activated CD38⁺ cells (Fig. 2e). In fact, the T_{EX} and T_{ML} do not positively associate with magnitude of response in Fig. 2e.

We have checked and the text reported correctly what is shown in Fig. 2E, such as T_{EX} is significantly correlated with IFN- γ only for CH (continuous line) and not CL (dashed line). We have checked all the correlations and provided additional ones utilising other T cell protein markers, which reinforce further the relationship between magnitude of response and specific T cell subsets (Supplementary Fig. 5).

4) The trajectory analysis is also confused me. It is unknown how the trajectories were grouped in two families (4 pseudotime lineages). It will be much better to show the pseudotime with all cells in fig4 before the authors separated them into two families (4 pseudotime lineages).

We have re-performed the trajectory analysis and removed the two families. We have added an additional analysis on all cells using Slingshot (Supplementary Fig. S6). We clarified the approach to choose trajectories in the CH (Fig. 4) and CL (Supp. Fig. 10) as well as in the monoclonal response identified in CL-MCRL (Fig. 5). In the revised results we have clarified the approach to identify trajectories which was based on the cluster connectivity as well as on the prior biological knowledge of the sampling time point. Roots were identified based on clusters comprising of cells sampled from the earliest time point and then trajectory was obtained by the identification of clusters that were fully-connected in the PAGA analysis. Please also note the responses to reviewers 1 and 2 in this regard.

5) Fig5f, how the cells were classified into two lineages since the overall pseudotime were not provided.

We provided pseudotime in the two lineages identified within this monoclonal response, plotting these values on the PAGA structure. In the revised manuscript, we have further clarified the methodology and added more panels to clarify the trajectory analysis. We have also added in Fig. 4A a graph representing the overall strategy for trajectory analysis. Of note, we have re-performed PAGA with the SCTransform matrix of gene expression values, and confirmed the major findings. In this revised analysis we used more stringent criteria and now all the trajectories are based on fully connected clusters in PAGA.

Minor concerns:

1) The garbled codes on figures (fig2a, fig2b, fig7c, fig7d, fig7e) and resolution of fig6 affect reading.

We have simplified the names of the trajectories, checked for consistency, and also removed Fig. 7.

2) fig3c, the colors in figure legend and box plot of TPINT and TMEM are inconsistent. Also a typo TPEX (should be TPINT)

We have corrected these typos.

3) The sample numbers described in manuscript are inconsistent with that in figures. e.g. identified in five CLs and seven CHs.

We have checked and we could not find any discrepancy. Perhaps this reviewer is referring to the number of subjects used in the functional study using ELISpot where we have analysed a total of 17 subjects (Fig. 1, Supplementary Table 1).

For the FACS analysis (Fig. 2), we have utilised a total of 12 subjects with multiple time points:

“We next investigated the phenotype of HCV-specific responses over the course of the infection, utilising dextramers to longitudinally study the phenotype of 20 epitope-specific CD8⁺ T cell responses that were identified in five CLs and seven CHs (Supplementary Table 4).”

While for the single cell data we have utilised 9 subjects (Fig. 3):

“In total, we analysed 1603 single cells from nine subjects (seven CHs and two CLs).”

4) “chronic progressor” was abbreviated as CH in most case, while sometimes abbreviated as Ch or even chronic. These abbreviation should be unified.

We thank this reviewer and have unified the abbreviations to Ch.

5) There is no information in Supplementary Fig. 2a.

We apologise for this issue. As above, we have noticed that in some computers several panels appeared empty. We have now transformed all our panels in Adobe Illustrator and we checked the result in both Apple and Windows systems.

References

1. Bull, R.A. *et al.* Transmitted/Founder Viruses Rapidly Escape from CD8⁺ T Cell Responses in Acute Hepatitis C Virus Infection. *J Virol* **89**, 5478-5490 (2015).
2. Goonetilleke, N. *et al.* The first T cell response to transmitted/founder virus contributes to the control of acute viremia in HIV-1 infection. *J Exp Med* **206**, 1253-1272 (2009).
3. Stryhn, A. *et al.* A Systematic, Unbiased Mapping of CD8(+) and CD4(+) T Cell Epitopes in Yellow Fever Vaccinees. *Front Immunol* **11**, 1836 (2020).
4. Lynn, R.C. *et al.* c-Jun overexpression in CAR T cells induces exhaustion resistance. *Nature* **576**, 293-300 (2019).
5. Bull, R.A. *et al.* Sequential bottlenecks drive viral evolution in early acute hepatitis C virus infection. *PLoS Pathog* **7**, e1002243 (2011).

6. Walker, M.R. *et al.* Clearance of hepatitis C virus is associated with early and potent but narrowly-directed, Envelope-specific antibodies. *Sci Rep* **9**, 13300 (2019).
7. Keck, Z.Y. *et al.* Broadly neutralizing antibodies from an individual that naturally cleared multiple hepatitis C virus infections uncover molecular determinants for E2 targeting and vaccine design. *PLoS Pathog* **15**, e1007772 (2019).
8. Osburn, W.O. *et al.* Clearance of hepatitis C infection is associated with the early appearance of broad neutralizing antibody responses. *Hepatology* **59**, 2140-2151 (2014).
9. Kinchen, V.J., Cox, A.L. & Bailey, J.R. Can Broadly Neutralizing Monoclonal Antibodies Lead to a Hepatitis C Virus Vaccine? *Trends Microbiol* **26**, 854-864 (2018).
10. Brasher, N.A. *et al.* B cell immunodominance in primary hepatitis C virus infection. *J Hepatol* **72**, 670-679 (2020).
11. Zheng, C. *et al.* Landscape of Infiltrating T Cells in Liver Cancer Revealed by Single-Cell Sequencing. *Cell* **169**, 1342-1356 e1316 (2017).
12. Qiu, Z., Chu, T.H. & Sheridan, B.S. TGF-beta: Many Paths to CD103(+) CD8 T Cell Residency. *Cells* **10** (2021).

REVIEWERS' COMMENTS

Reviewer #1 (Remarks to the Author):

The authors have carefully addressed my comments.

Reviewer #2 (Remarks to the Author):

Summary of revisions

Cai et al present thorough and robust revisions of their originally submitted manuscript. Specifically, the authors have addressed the major comments that arose initially including an assessment of HCV viral diversity, an appropriate comment in the discussion regarding humoral responses, revised application of methodology for integration of scRNAseq data, and revised methodology and justification for pseudotemporal inference. The revised manuscript is much improved and advances the understanding of human CD8+ T cell exhaustion over time in chronic viral infection. There are only minor comments remaining from this reviewer.

Minor comment #1

The authors present analysis of viral diversity in Figure 1d and Supplementary Figure 1b using Shannon entropy. These results revealed no differences in viral diversity between chronically infected patient and patients that clear HCV, greatly strengthening their conclusions regarding early CD8+ T cell exhaustion. While the measures of viral diversity address the initial question, further support of the authors' hypothesis could be derived from analysis of the emergence (or disappearance) of specific viral subpopulations over time. The authors' hypothesis that robust early CD8+ T cell responses lead to viral escape could be validated by identifying viral subpopulations that disappear initially and new viral quasi-species that emerge over time. This is probably beyond the scope of the manuscript but could be mentioned in the discussion as a future direction.

Minor comment #2

In figure panels such as Figure 1i where the authors are including lines of fit, it would be ideal to include confidence intervals around such lines.

Minor comment #3

In the presentation of the pseudotemporal analyses in the results, there may be some confusion based on the language used between the literal timepoint (i.e. early post-infection, late post-infection) from which the cells are derived and the cell state. For example, in Figure 3 the results sections states that an "...increased proportion of Tex cells in cluster 4 and 5 which were predominantly early-stage cells and with high IFN-g response." This may be confusing to some readers because the cells are early-stage with regards to disease state, but are NOT early-stage with regards to T cell differentiation (they are terminally differentiated). Thus, the authors should carefully distinguish between disease state and T cell differentiation stage in this section.

Reviewer #3 (Remarks to the Author):

The manuscript is much improved. Although I still think the manuscript is statistically imperfect, the manuscript has a good story and logically reasonable. Therefore, I support the publication of this manuscript.

REVIEWERS' COMMENTS

Reviewer #1 (Remarks to the Author):

The authors have carefully addressed my comments.

Reviewer #2 (Remarks to the Author):

Summary of revisions

Cai et al present thorough and robust revisions of their originally submitted manuscript. Specifically, the authors have addressed the major comments that arose initially including an assessment of HCV viral diversity, an appropriate comment in the discussion regarding humoral responses, revised application of methodology for integration of scRNAseq data, and revised methodology and justification for pseudotemporal inference. The revised manuscript is much improved and advances the understanding of human CD8+ T cell exhaustion over time in chronic viral infection. There are only minor comments remaining from this reviewer.

Minor comment #1

The authors present analysis of viral diversity in Figure 1d and Supplementary Figure 1b using Shannon entropy. These results revealed no differences in viral diversity between chronically infected patient and patients that clear HCV, greatly strengthening their conclusions regarding early CD8+ T cell exhaustion. While the measures of viral diversity address the initial question, further support of the authors' hypothesis could be derived from analysis of the emergence (or disappearance) of specific viral subpopulations over time. The authors' hypothesis that robust early CD8+ T cell responses lead to viral escape could be validated by identifying viral subpopulations that disappear initially and new viral quasi-species that emerge over time. This is probably beyond the scope of the manuscript but could be mentioned in the discussion as a future direction.

We have now specifically added a sentence in the revised manuscript about the emergence of novel escape viral variants in the viral populations. This study has been published by our group in Bull et al J. Virol 2015 and Bull et al PLOS Pathogens 2011 where we studied the kinetics of viral variants that characterise the early stage of infection of HCV. These previous studies included a proportion of the subjects that have been used in this manuscript. Specifically, Table 2 in Bull et al J Virol 2015 reports examples from 4 patients where specific amino mutations were identified following infection with the transmitted found virus and these include within epitope mutations, which were validated using IFN-g ELISPOT.

"As previously shown, we have demonstrated that a genetic bottleneck in viral populations occur at around 3 months post-infection, where the transmitted founder viruses are replaced with novel variants that dominate the populations and carry amino acid changes across the viral genome, including within HLA-I restricted epitopes^{16, 17}. These previous findings from

our group support the conclusion that the magnitude of CD8⁺ T cell responses influence viral evolution in acute phase of primary HCV infection and onset of escape variants."

Minor comment #2

In figure panels such as Figure 1i where the authors are including lines of fit, it would be ideal to include confidence intervals around such lines.

We have added the confidence intervals for the linear regression.

Minor comment #3

In the presentation of the pseudotemporal analyses in the results, there may be some confusion based on the language used between the literal timepoint (i.e. early post-infection, late post-infection) from which the cells are derived and the cell state. For example, in Figure 3 the results sections states that an "...increased proportion of Tex cells in cluster 4 and 5 which were predominantly early-stage cells and with high IFN-g response." This may be confusing to some readers because the cells are early-stage with regards to disease state, but are NOT early-stage with regards to T cell differentiation (they are terminally differentiated). Thus, the authors should carefully distinguish between disease state and T cell differentiation stage in this section.

We have now added a statement clarifying these two concepts.

Early stage "During the early-stage (defined as early phase of infection, i.e., the first 120 days post-infection (DPI))".

We have also modified the sentence as

"The distribution of T cell phenotypes within each cluster revealed an increased proportion of terminally differentiated T_{EX} cells in clusters 4 and 5 which were predominantly found from samples in early-stage of infection and with high IFN- γ response (Fig. 3c, Supplementary Fig. 6a)."

Reviewer #3 (Remarks to the Author):

The manuscript is much improved. Although I still think the manuscript is statistically imperfect, the manuscript has a good story and logically reasonable. Therefore, I support the publication of this manuscript.